# Multi-state design of flexible proteins predicts sequences optimal for conformational change

**Marion F. Sauer** [1,2], **Alexander M. Sevy** [1,2], **James E. Crowe, Jr.** [2,3,4], **Jens Meiler** [1,5]*

**1** Center for Structural Biology, Vanderbilt University, Nashville, Tennessee, United States of America,
**2** Vanderbilt Vaccine Center, Vanderbilt University Medical Center, Nashville, Tennessee, United States of
America, **3** Department of Pediatrics, Vanderbilt University Medical Center, Nashville, Tennessee, United
States of America, **4** Department of Pathology, Microbiology and Immunology, Vanderbilt University Medical
Center, Nashville, Tennessee, United States of America, **5** Department of Chemistry, Vanderbilt University,
Nashville, Tennessee, United States of America

* jens@meilerlab.org

SWEDEN

**Data Availability Statement:** Analysis of all data
are contained within the manuscript and its
Supporting Information files. All PDB files are
available from the Protein Data Bank (https://www.
rcsb.org) and accession numbers are listed in

## Abstract

Computational protein design of an ensemble of conformations for one protein–*i.e.*, multi-state
design–determines the side chain identity by optimizing the energetic contributions of that side
chain in each of the backbone conformations. Sampling the resulting large sequence-structure
search space limits the number of conformations and the size of proteins in multi-state design
algorithms. Here, we demonstrated that the REstrained CONvergence (RECON) algorithm
can simultaneously evaluate the sequence of large proteins that undergo substantial conformational changes. Simultaneous optimization of side chain conformations across all conformations increased sequence conservation when compared to single-state designs in all
cases. More importantly, the sequence space sampled by RECON MSD resembled the evolutionary sequence space of flexible proteins, particularly when confined to predicting the mutational preferences of limited common ancestral descent, such as in the case of influenza type
A hemagglutinin. Additionally, we found that sequence positions which require substantial
changes in their local environment across an ensemble of conformations are more likely to be
conserved. These increased conservation rates are better captured by RECON MSD over
multiple conformations and thus multiple local residue environments during design. To quantify this rewiring of contacts at a certain position in sequence and structure, we introduced a
new metric designated 'contact proximity deviation' that enumerates contact map changes.
This measure allows mapping of global conformational changes into local side chain proximity
adjustments, a property not captured by traditional global similarity metrics such as RMSD or
local similarity metrics such as changes in $\varphi$ and $\psi$ angles.

## Author summary

Multi-state design can be used to engineer proteins that need to exist in multiple conformations or that bind to multiple partner molecules. In essence, multi-state design selects a
compromise of protein sequences that allow for an ensemble of protein conformations, or
states, associated with a particular biological function. In this paper, we used the

Table 1. Software used to run Rosetta RECON multi-state design and single state design are available for download from the Rosetta Commons, https://github.com/RosettaCommons/main.

**Funding:** M.F.S., A.M.S., J.E.C., and J.M. received funding through the National Institute of Health U19 AI117905 grant for Structure Based Design of Antibodies and Vaccines (http://grantome.com/grant/NIH/U19-AI117905-01). The funders had no role in study design, data collection and analysis, decision to publish, or preparation of the manuscript.

**Competing interests:** The authors have declared that no competing interests exist.

REstrained CONvergence (RECON) algorithm with Rosetta to show that multi-state design of flexible proteins predicts sequences optimal for conformational change, mimicking mutation preferences sampled in evolution. Modeling optimal local side chain physicochemical environments within an ensemble selected significantly more native-like sequences than selections performed when all conformations states are designed independently. This outcome was particularly true for amino acids whose local side chain environment change between conformations. To quantify such contact map changes, we introduced a novel metric to show that sequence conservation is dependent on protein flexibility, *i.e.*, changes in local side chain environments between stated limit the space of tolerated mutations. Additionally, such positions in sequence and structure are more likely to be energetically frustrated, at least in some states. Importantly, we showed that multi-state design over an ensemble of conformations (space) can explore evolutionary tolerated sequence space (time), thus enabling RECON to not only design proteins that require multiple states for function but also predict mutations that might be tolerated in native proteins but have not yet been explored by evolution. The latter aspect can be important to anticipate escape mutations, for example in pathogens or oncoproteins.

## Introduction

Computational protein design solves the so-called 'inverse folding problem' by identifying an amino acid sequence that is compatible with a given protein structure, *i.e.*, backbone conformation and possibly interactions with partner biomolecules. This approach allows for the molecule to conduct its function in this single state. Protein function, however, often relies on the transition between multiple conformations–a protein must be thermodynamically stable in multiple conformations before it is capable of achieving a defined function. Thus, for a protein to conserve its function, we hypothesized that the conservation of protein flexibility limits the protein's sequence space to be consistent with the conformational changes needed for function. Determining functionally relevant sequence tolerance, or rather, the set of amino acid sequences that are allowable given a protein's function, therefore depends on identifying the set of amino acid sequences that is stable in each of the conformations needed. Testing this hypothesis is complicated, as typically not all functionally relevant conformations have been determined experimentally. The picture gets even more complicated if we look not only at functionally relevant conformations that are by definition local free energy minima (*i.e.*, thermodynamics) but also include an analysis of the height of barriers connecting these states that determine the kinetics of interconversion.

Humphris-Narayanan and colleagues demonstrated that prediction of mutation preferences of HIV-1 protease and HIV-1 reverse transcriptase was improved up to 25% when structural ensembles were included during protein design, as opposed to design of a single conformation [1, 2]. The structural ensembles used for this approach were generated by reverting all structural side chains to a consensus sequence and employing ROSETTA Backrub to introduce small local rotation about the $C_\alpha$-$C_\alpha$ axis of each of three-residue segments while maintaining ideal bond length, angle, and the starting $\chi1$ angle [3, 4] at sites distributed throughout the protein known to acquire mutations. Next, protein design was performed on each backbone within the ensemble to select for mutations sequence that contributed to increased protein fold, dimer, and peptide stability. which was calculated as the lowest weighted sum of energy scores. They found that the substitution frequency of the consensus sequence, or profile, of the backbone ensemble better corresponded to the mutation frequencies observed within the Stanford HIV-1

Database [5] than the substitution frequencies obtained from design of an individual conformation. Additionally, they showed that the sequence profiles acquired with their Backrub ensembles were similar to the sequence profiles attained using an ensemble of experimentally-derived structural models. With such results, this approach succeeded in showing that representation of conformational plasticity during protein design better mimicked the mutational tolerances of HIV-1 protease and reverse transcriptase. The authors attribute this finding to the requirement of small backbone changes to accommodate mutations from the starting sequence.

For certain proteins, sub-Angstrom perturbations of the peptide backbone are insufficient to represent the conformational space consistent with their function. In the case of ubiquitin, Friedland and colleagues also used ROSETTA Backrub to generate ubiquitin ensembles, but unlike in the previously mentioned method, they randomly inserted local rotations about the $C_\alpha$-$C_\alpha$ axis of two to twelve residues to diversify the conformational space of the generated ensembles. They then culled any generated models which disagreed with nuclear magnetic resonance (NMR) residual dipolar coupling constraints (RDCs), thus generating ubiquitin ensembles more similar to native-state solution dynamics [6]. Using these RDC-constrained ensembles for design, they demonstrated that the mutation profiles obtained from the collection of individually-designed poses were more consistent with sequences within the ubiquitin family. In combination with the aforementioned study, these approaches demonstrated that a requirement for protein flexibility of a native-state ensemble substantially dictates the sequence space available for evolution.

A limitation of these approaches, however, is the assumption that the tolerated sequence space for a conformationally flexible protein can be determined by integrating over each single-state design (SSD) profile of each conformation within an ensemble, *i.e.*, enumerating the most energetically favorable amino acid for each position and each conformation. However, the most energetically favorable amino acid, as determined by the aforementioned methods, may be the most energetically favorable for one or more single conformations, but may not be energetically tolerable at the same position in another conformation. For instance, in a certain conformation, the energetically most favorable amino acid might be the only allowed amino acid, with all others prohibited (imagine a tiny space where only glycine fits). At the same position in other conformations, there may be acceptable alternatives with more energetically favorable scores, but those residues could not be tolerated as an acceptable mutation in the aforementioned more constrained position. Thus, we hypothesized that multi-state design (MSD) over all conformations relevant for function will yield a more accurate representation of the biologically relevant sequence space compatible with function.

Using a pre-defined scoring function, positive-state MSD approaches rank the stability of a sequence as the average stability when threaded over each state across the ensemble [7]. For most MSD approaches, replacement of the starting, or native, sequence occurs only if the mutant improves with respect to the starting sequences. The Best Max-Marginal First (BMMF) algorithm was used to demonstrate that the MSD of 16 unique calmodulin-substrate complexes increased the similarity of the designed calmodulin binding site to evolutionary sequence profiles by two-fold, and increased in native sequence recovery from 52.5% for single-state designs to 80% for the 16-state design scenario [8]. Challenges for applying MSD methods like the BMMF algorithm, however, are the efficiency of the search algorithm, large memory requirements, and extended computational time needed. MSD methods up to now have been limited to designing a small number of amino acid positions across all states, with the largest number of simultaneously evaluated design positions being 27 designed positions across 60 states using the MSD FASTER algorithm [8–12].

We sought to study large proteins that undergo conformational rearrangements that include domain or hinge displacements of greater than a few Å in root mean square distance (RMSD).

We expected that the tolerated sequence space must be restricted in some regions to allow for substantial 'rewiring' of contact networks when transitioning from one state to another. The tolerated sequence space of these types of conformational changes is not limited to local regions, such as protein-protein interfaces, but instead distributed over the entire amino acid sequence. Thus, an MSD approach that seeks to explore such sequence spaces needs to include the entire protein. The REstrained CONvergence (RECON) algorithm was used previously to estimate the sequence tolerance within protein-protein interfaces. However, already at that time this approach proved to be more computationally efficient than the generic ROSETTA MSD algorithm [12]. With the addition of a message-passage interface (MPI), RECON MSD can combine the single-state design (SSD) efficiency of evaluating the sequence tolerance of a full-length protein with the MSD capability of evaluating the fitness function of a sequence across multiple conformations [13].

Highly flexible viral glycoproteins, such as the influenza A hemagglutinin protein and its stem domain (HA2), undergo conformational rearrangements of greater than 30 Å and have been shown to be conserved in sequence greater than 90% percent across subtypes [14]. For other highly flexible proteins, such as calmodulin, kinases, and voltage-gated sensory channels, regions known to mediate conformational change can be conserved up to 100% across phylogenies, suggesting that a limited set of sequences is suitable for select conformation transitions [15–17]. Here we use the ROSETTA RECON MSD algorithm to demonstrate that the sequence space consistent with all experimentally determined conformations of a protein approximates sequence profiles observed in evolution.

## Results

It is our aim to demonstrate that simultaneous evaluation of sequence space across an ensemble of conformations improves the correspondence of the designed sequences to an evolutionary sequence profile by considering the constraints that local and global protein flexibility impose on amino acid identity and rotamer placement. For this benchmark, we perform RECON MSD and compare the designed profiles to SSD and PSI-BLAST profiles to quantify the similarity between designs and evolutionary profiles (Fig 1).

### Compilation of a benchmark set of eight proteins

We selected proteins with multiple known conformations of identical sequence from the PDBFlex database [18]. The benchmark included eight proteins, requiring that each benchmark case have at least two published conformations with an RMSD greater than 5 Å, and an identical sequence greater than 100 amino acids in length (Table 1). We omitted duplicate conformations, which we define as conformations with and RMSD of less than 0.5 Å, to avoid design bias towards similar conformations. In addition, we used a resolution cutoff of 5 Å with the requirement that greater than 75% of the included models within each design ensemble were determined at a resolution of better than 3 Å. We also omitted any models with longer sequence gaps or missing densities. For structural models with chain breaks that had missing density for only one or two consecutive residues (PDB IDs 1OK8, 3C5X, and 3C6E of the dengue virus E protein monomer) we added the missing densities with the Rosetta loop modeling application [19]. All structural models were gently relaxed with a restraint to start coordinates to remove any energetic frustrations frequent in models derived from low-resolution experimental structures.

### Metrics to measure amplitudes of local and global conformational change

Quantification of protein flexibility commonly relies on the structural comparison of two structural models, whether that be through the similarity of equivalent atoms in three-

**A**

**B**

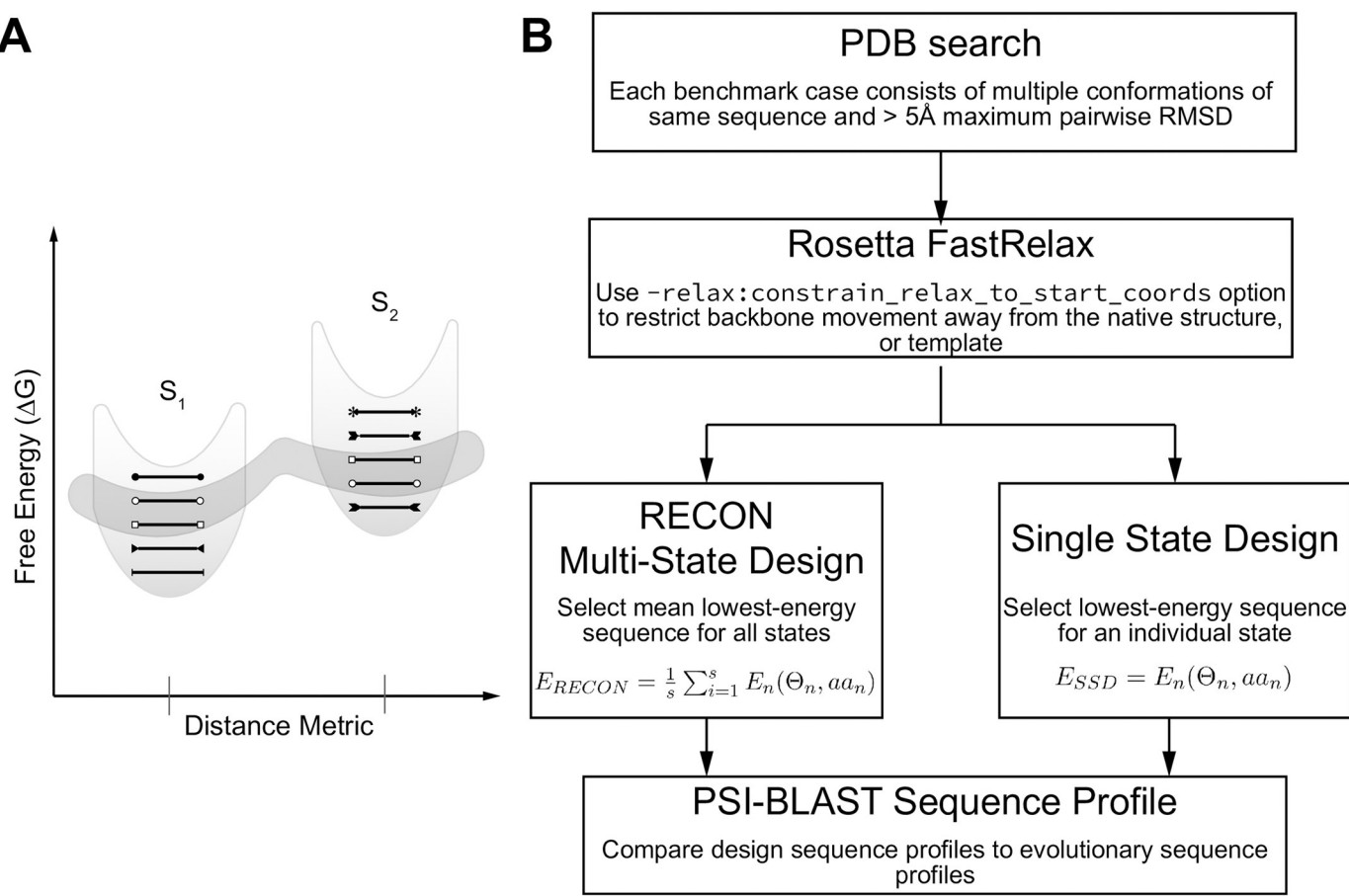

**Fig 1. Graphical representation of hypothesis and experimental design.** (A) Schematic of sequence space and the impact of flexibility on sequence tolerance. $S_1$ and $S_2$ represent two unique conformations of the same residue length separated by some RMSD that populate two local energy minima. Black lines with end caps represent unique sequences that are energetically most favorable for a single conformation. The dark shaded area encircles sequences that are energetically favorable for both conformations. Here we illustrate that by using multiple conformations during protein design, we identify sequences that are energetically suitable for conformational flexibility, yet are not necessarily the most stable sequence for any given conformation. Additionally, the requirement to adopt multiple conformations constrains the number of suitable sequences (B) Flow chart of benchmark design.

dimensional space, calculated as root mean square distance (RMSD), or by the similarity of equivalent φ and ψ backbone dihedral angles, calculated as root mean square deviation ($RMSD_{da}$) of a $C_\alpha$ atom. [20]. RMSD is used frequently as a global metric used to describe the overall similarity of two conformations of the same protein and has been a powerful metric to quantify overall structural similarity. $RMSD_{da}$, on the other hand, is used to describe local backbone displacements and is well-established, for example, to compare loop conformations. The disadvantage of both metrics is that they do not capture whether or not a particular residue is reconfigured in its interactions with neighboring amino acids. However, we hypothesize that such a metric of local rewiring driven by a global conformational space will best correlate with restrictions in sequence space introduced through conformational flexibility. Thus, we settled on three metrics that capture the structural *dis*similarity of a protein ensemble in terms of its maximum global structural dissimilarity, local backbone dissimilarity, or contact map dissimilarity: 1) The maximum pairwise RMSD of all atom coordinates of two superimposed structures within a set of *n* superimposed structures was used as a metric to describe the maximal global conformational change an ensemble undergoes (Fig 2A). To allow for comparison of RMSD values between benchmark cases that involve proteins of different size, we used

**Table 1. Proteins used in conformation-dependent sequence tolerance benchmark.**

| PROTEIN | PDB ID OF STATES USED WITHIN EACH ENSEMBLE | DETERMINATION METHOD | DESIGNED POSITIONS | AVERAGE PAIRWISE RMSD OF DESIGNED STATES (Å) | LARGEST PAIRWISE RMSD (Å) |
|---|---|---|---|---|---|
| 5´-NUCLEOTIDASE | 1HPU | X-ray | 523 | 5.18 ± 1.12 | 9.20 |
| | 1OI8 | X-ray | | | |
| | 1IOD, chain A | X-ray | | | |
| | 1OID, chain B | X-ray | | | |
| | 4WWL | X-ray | | | |
| ADENYLATE KINASE | 1AKE | X-ray | 214 | 7.19 | 7.19 |
| | 4AKE | X-ray | | | |
| CAGL | 3ZCJ | X-ray | 169 | 27.0 ± 22.2 | 39.9 |
| | 4CII | X-ray | | | |
| | 4YVM | X-ray | | | |
| CALMODULIN | 1A29 | X-ray | 139 | 10.4 ± 2.44 | 17.9 |
| | 1CFC | NMR | | | |
| | 1CFD | NMR | | | |
| | 1CFF | NMR | | | |
| | 1CKK | NMR | | | |
| | 1CLL | X-ray | | | |
| | 1CM1 | X-ray | | | |
| | 1CM4 | X-ray | | | |
| | 1G4Y | X-ray | | | |
| | 1LIN | X-ray | | | |
| | 1MUX | NMR | | | |
| | 1NIW | X-ray | | | |
| | 1NWD | NMR | | | |
| | 2F2P | X-ray | | | |
| | 2N8J | NMR | | | |
| | 2WEL | X-ray | | | |
| | 3EWT | X-ray | | | |
| | 3EWV | X-ray | | | |
| | 4DJC | X-ray | | | |
| | 4HEX | X-ray | | | |
| DENGUE VIRUS ENVELOPE PROTEIN (MONOMER) | 1OAN | X-ray | 394 | 6.63 ± 2.89 | 13.3 |
| | 1OK8 | X-ray | | | |
| | 3C5X | X-ray | | | |
| | 3C6E | X-ray | | | |
| | 3J27 | Cryo-EM | | | |
| | 3J2P | Cryo-EM | | | |
| GROEL SUBUNIT | 1AON, chain A | X-ray | 523 | 9.06 ± 1.13 | 13.1 |
| | 1AON, chain N | X-ray | | | |
| | 2C7E | Cryo-EM | | | |
| | 3WVL | X-ray | | | |
| | 4AB3 | Cryo-EM | | | |
| | 4KI8 | X-ray | | | |
| INFLUENZA HEMAGGLUTININ STEM (TRIMER) | 1QU1 | X-ray | 344 | 23.7 ± 17.4 | 34.9 |
| | 1HTM | X-ray | | | |
| | 2HMG | X-ray | | | |
| | 3EYM | X-ray | | | |

*(Continued)*

**Table 1.** (Continued)

| PROTEIN | PDB ID OF STATES USED WITHIN EACH ENSEMBLE | DETERMINATION METHOD | DESIGNED POSITIONS | AVERAGE PAIRWISE RMSD OF DESIGNED STATES | LARGEST PAIRWISE RMSD |
|---|---|---|---|---|---|
| | | | | (Å) | (Å) |
| RESPIRATORY SYNCYTIAL VIRUS FUSION PROTEIN (TRIMER) | 3RKI | X-ray | 1252 | 29.9 ± 22.0 | 46.2 |
| | 3RRR | X-ray | | | |
| | 4MMS | X-ray | | | |
| | 4ZYP | X-ray | | | |

For a complete description of Protein Data Bank (PDB) identification and sequence information included in the benchmark, see S1 Table.

RMSD100, a RMSD value normalized to protein of length 100 amino acids. [21] 2) Residue $\phi$ and $\varphi$ RMSD$_{da}$ was used as a local metric of similarity (Fig 2B). This metric will directly identify hinge regions between moving domains. 3) Lastly, we designed a metric that captures changes in the contact map computed as $C_\beta$–$C_\beta$ distance variation. This metric captures local changes in the environment of a residue by including non-local tertiary contacts in the analysis. Thus, it is designed to capture the local and global changes of the physicochemical environment of a residue and thus defines which amino acids are tolerated in a certain position (Fig 2C and 2D). For a complete description of each metric, see Methods.

## RECON MSD samples sequence profiles that are more similar to evolutionary observed sequence profiles when compared to SSD

We first examined the correspondence of native sequence recovery determined by MSD versus SSD designed sequences to conservation rates within natural homologues. To accomplish this goal, we performed either RECON MSD or SSD on each set of protein conformations, allowing for the substitution of the native residue to all twenty amino acids and ignoring the presence of any disulfide bonds present in the native model. Designed sequence profiles of each conformation were generated using ten designed sequences, which were selected from either the ten lowest-scoring designed ensembles, as in the case of RECON MSD, or conformations for SSD. The mean total score of all conformations designed during the same RECON MSD run was used to sort and select the ten lowest-scoring designed ensembles. From the ten mean lowest-scoring ensembles, the set of ten models of each conformation was used for analysis. For SSD, each conformation was designed independently, and therefore the ten lowest-scoring models of each conformation were used for analysis. Therefore, the distinction between RECON MSD and SSD selected models rested in whether or not each of the ten selected models of each conformation where evaluated for design collectively as an ensemble or not. To compare the designed sequence profiles to that of the sequence diversity in natural homologues, the native sequence was used as the PSI-BLAST query sequence to generate PSI-BLAST profiles for each protein. Native sequence recovery was calculated as the mean percentage of conservation of the starting, or native, sequence for all designed positions, and for consistency, we term the percentage of the query sequence used to generate each PSI-BLAST profile as the percent native sequence recovery.

Simultaneously sampling across multiple conformations significantly restricted sequence sampling, or in other words, was more likely to conserve the native sequence, where RECON MSD had total native sequence recovery of 87.8 ± 4.5% versus SSD with 48.9 ± 11.1% native sequence recovery (Fig 3A). In contrast, PSI-BLAST profiles had a native sequence recovery of 82.11 ± 11.2%. Qualitatively, the PSI-BLAST profiles were much more similar to the predicted

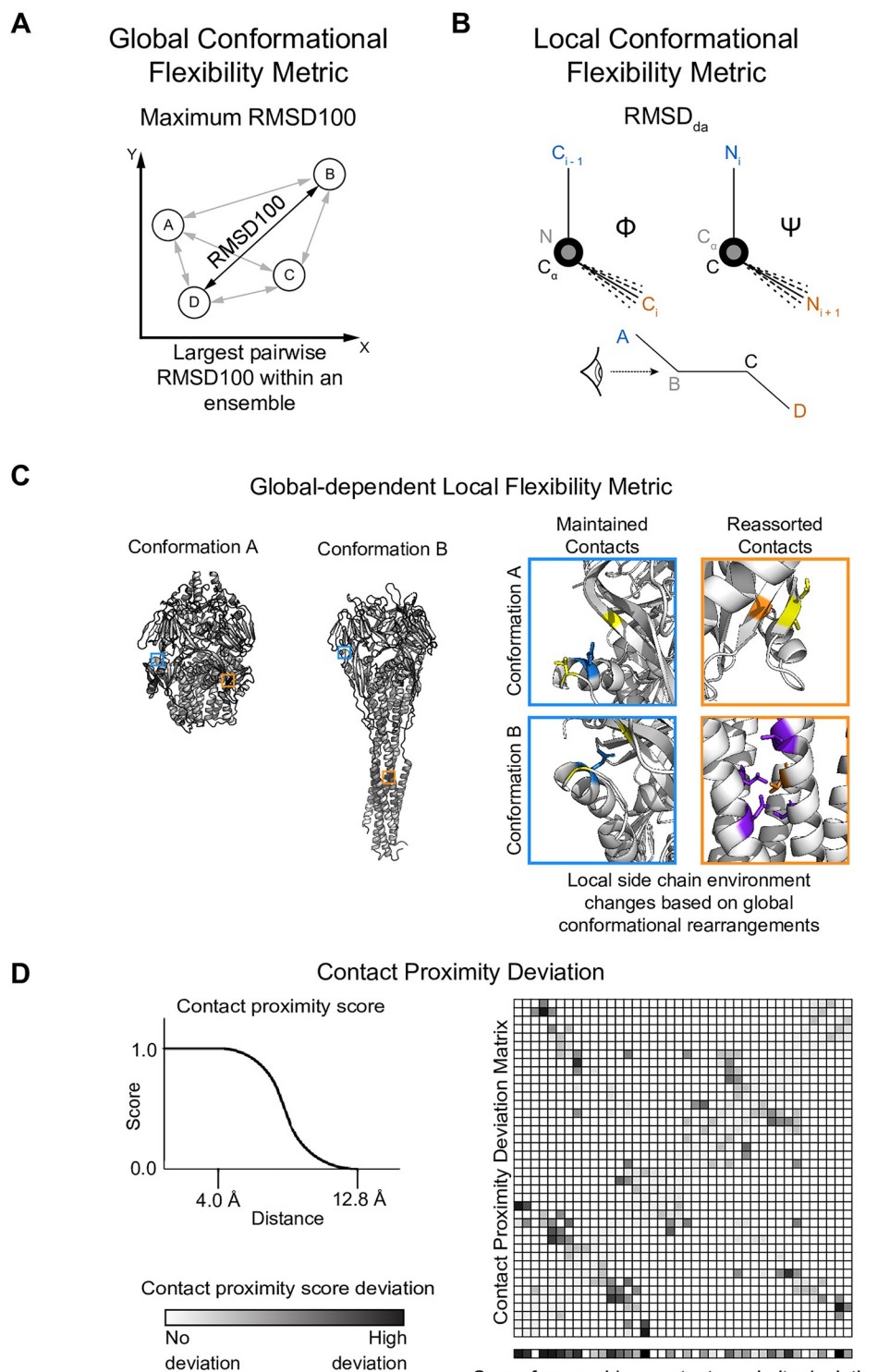

**Fig 2. Metrics used to quantify conformational flexibility.** (A) Illustration of maximum RMSD100, the metric used to quantify large-scale, or global, conformational flexibility. For simplicity, we only represent RMSD on a two-dimensional plane, where the x and y axes represent the difference in distance of cartesian space if two conformations were superimposed onto the same coordinate system. Each protein conformation of identical sequence is represented as a circle, and is separated by some distance vector evaluated as the RMSD100 of two conformations. The maximum RMSD100 describes the greatest pairwise RMSD100 within an ensemble. (B) Illustration of dihedral angle $\phi$ and $\varphi$ variation used to calculate dihedral angle RMSD (RMSD$_{da}$). Orientation of atoms is color-coded and corresponds to

the diagram drawn at the bottom of the panel. $RMSD_{da}$ is illustrated as the range of dotted lines, corresponding to the deviation in relative orientation of the third and fourth atoms. (C) Explanation of contact proximity deviation. Two conformations of the same protein are depicted in the left, with two residues, outlined in cyan or orange, shown in their respective positions. These two residues are magnified (top right) in their local side chain environment in Conformation A on the top and Conformation B on the bottom. Contact residues in Conformation A are colored yellow. If the same contacts are maintained in Conformation B, contact residues remain colored yellow in the bottom two boxes. If new contacts are made, contact residues are colored in purple. Even though the cyan residue changes slightly in its relative orientation between conformations, the same contacts are maintained so that the degree of conformational flexibility is relatively low in comparison to the heptad trimer refolding, and would have a low contact proximity deviation score. In contrast, the orange residue completely rearranges its local side chain contacts between conformations as a result of the large conformational rearrangement, and would have a high contact proximity deviation score. (D) Explanation of contact proximity deviation. We assigned a score to each $C_\beta$–$C_\beta$ distance by applying a soft-bounded, continuously differentiable function that accounts for the proximity of two side chains and approximates the likelihood of two side chains forming a contact, illustrated in the top left of Panel D. We then calculated the deviation of each $C_\beta$–$C_\beta$ distance across an ensemble as shown in the matrix, with low deviation scores in white and high scores in black. The contact proximity deviation score represents the sum of all $C_\beta$–$C_\beta$ proximity deviations a single residue undergoes within an ensemble, as shown in the bottom row separated from the matrix.

sequence tolerance of RECON MSD compared to SSD, yet a Mann-Whitney U test [22] indicated a significant difference of mean native sequence recovery of either design protocol compared to PSI-BLAST sequence tolerances, with a significance of $p = 0.0029$ for RECON MSD and $p < 0.00001$ for SSD. Total sequence recovery is a coarse approximation of sequence

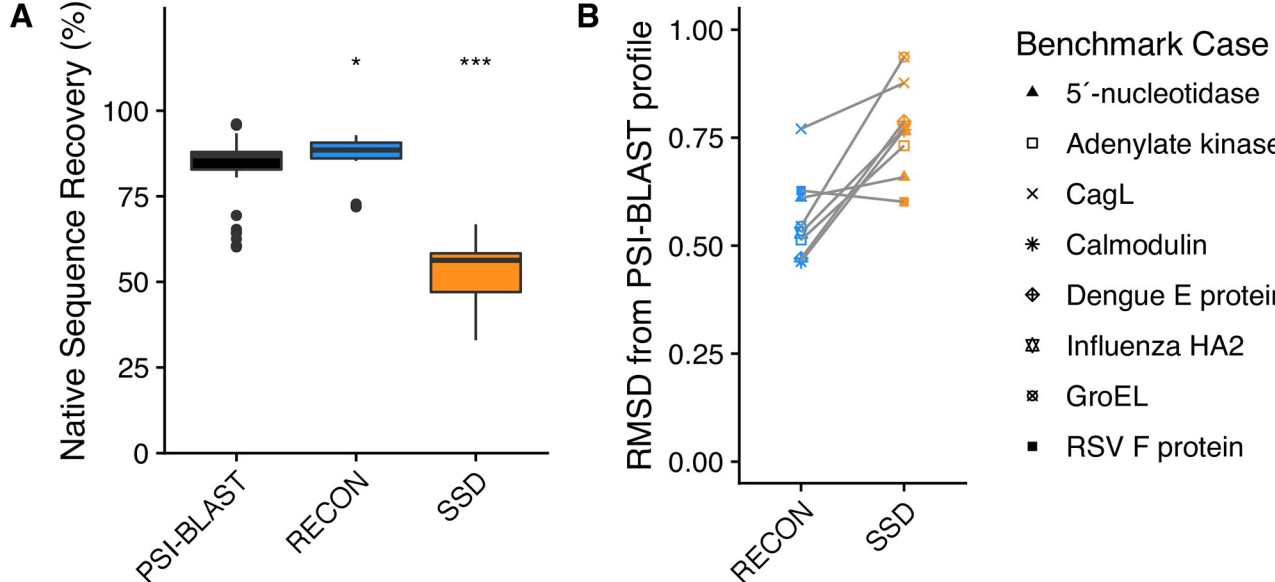

**Fig 3. Design native sequence recovery and mutation profile variability comparisons to PSI-BLAST profiles.** (A) Comparison of total native sequence recovery of relaxed and unminimized RECON MSD and SSD designs to PSI-BLAST sequence profiles generated using the native sequence. For this figure and all subsequent boxplots, shaded regions of each box plot denote values within the first and third quartiles (interquartile range, or *IQR*), with the median indicated as a solid line and whiskers representing values ± 1.5 × *IQR*. Outliers are represented as dots. Asterisks indicate the significance of difference of means of each design in comparison to the PSI-BLAST profile, with a *z*-test *p*-value < 0.01 represented by one asterisk, and a *p*-value < 0.00001 by three asterisks. The *p*-value provided in this figure and all subsequent figures represents a two-sided, 95% confidence interval. (B) Mutation frequency root mean square deviations of designs in comparison to a PSI-BLAST profile. The y-axis values represent the root mean square

deviation (RMSD) of mutation profiles for each designed residue in relation to a PSI-BLAST profile, represented as: $y = \sqrt{\frac{\sum_{i=1}^{n}\sum_{j=1}^{20}(aa_{PSI-BLAST}-aa_{Design})^2}{n}}$ where $aa_j$ represents the frequency of an amino acid observed at position *i* for each of all twenty amino acids (*j*), and *y* is the sum of all *i* differences for all amino acids within a protein of length *n* residues. A y-value of 0 would indicate that the design profile is identical to the PSI-BLAST profile, and an increase in y-value indicates the root mean square deviation of the sequence profile for each residue is more dissimilar to a PSI-BLAST profile.

similarity, and fails to determine if the designed sequence profiles are sampling similar mutation preferences as observed in evolution. Therefore, we calculated an average total deviation score of each observed position-specific mutation profile to the corresponding PSI-BLAST profile of each protein (Fig 3B and S1 Fig), which depicts the root mean square deviation of all aligned positions' profiles between a natural homologues' mutational preferences and those predicted by design. We found that in seven out of eight cases, a RECON MSD mutation profile resembled its corresponding PSI-BLAST profile more closely than the SSD mutation profile, as the root mean square deviation was lower for comparisons between PSI-BLAST profiles and RECON MSD profiles than that of between SSD profiles.

## RECON MSD underestimates amino acid exchangeability, but samples a more evolutionarily relevant sequence space than SSD

Although RECON MSD more closely resembled PSI-BLAST sequence profiles on a per-case basis, we wanted to identify trends in sequence sampling in relation to the PSI-BLAST profiles to highlight design-sampling biases. This task was achieved by calculating the frequency an amino acid is conserved or mutated to another residue, or, the mean amino acid substitution frequency. In general, RECON MSD is more likely to conserve a native amino acid compared to a PSI-BLAST profile, whereas SSD is much more likely to replace the native amino acid (Fig 4A and S2 Fig). We examined amino acid exchangeability as the frequency of exchanging a native for a non-native amino acid. On average, PSI-BLAST profiles exchanged a native for non-native amino acid 1.32 ± 0.03% of the time, versus 0.77 ± 0.02% for RECON MSD and 2.45 ± 0.07% for SSD (Fig 4B). Additionally, we compared the average difference of exchangeability for each residue as observed in the PSI-BLAST profiles versus either RECON MSD or SSD and found that RECON MSD average exchangeability rates of each residue are more similar to PSI-BLAST values than SSD (Fig 4C). With the exception of phenylalanine or tyrosine, the difference between exchangeability rates for residues with larger side chains diminishes for RECON MSD, but becomes more exaggerated for SSD, as compared to observed mutation rates in evolution. This finding suggests that the inclusion of multiple conformations during design encourages better placement of bulky side chains, albeit with conservative placement. However, when comparing the linear regression model of individual exchangeability rates of either RECON MSD or SSD to PSI-BLAST rates, both designs were roughly equally dissimilar to PSI-BLAST exchangeability rates, with RECON MSD having a correlation coefficient of $r = 0.35$ and SSD with $r = 0.64$ (S3 Fig). Given that exchangeability rates were not normally distributed, a Kendall $\tau_\beta$ rank correlation coefficient [23] was computed to measure the ordinal association of design and PSI-BLAST amino acid exchangeability rates, where a coefficient of $\tau_\beta = 0$ would indicate that the amino acid exchangeability rates are identical. We found RECON MSD to have a $\tau_\beta = 0.283$, $p \leq 2.22 \times 10^{-16}$ versus $\tau_\beta = 0.372$, $p \leq 2.22 \times 10^{-16}$ for SSD when measured for its association to PSI-BLAST amino acid exchangeability rates. In addition, we compared the difference in exchangeability rates between design and PSI-BLAST by calculating the ratio of transformed exchangeability rates ($e$) as $\frac{e_{design} + 0.00001}{e_{PSI-BLAST} + 0.00001}$ to avoid division by zero, where an individual exchangeability rate would be equivalent between design and PSI-BLAST if $e_{ratio} = 1$. We found that for RECON MSD the mean $e_{ratio} = 2.49$ and for SSD the mean $e_{ratio} = 22.7$. A Mann-Whitney U test of matched individual ratios found a significant difference between RECON MSD and SSD exchangeability rate ratios to PSI-BLAST exchangeability rates, with $p < 0.0001$. Taken together, RECON MSD is sampling individual mutation preferences significantly more closely to that observed in evolution than SSD.

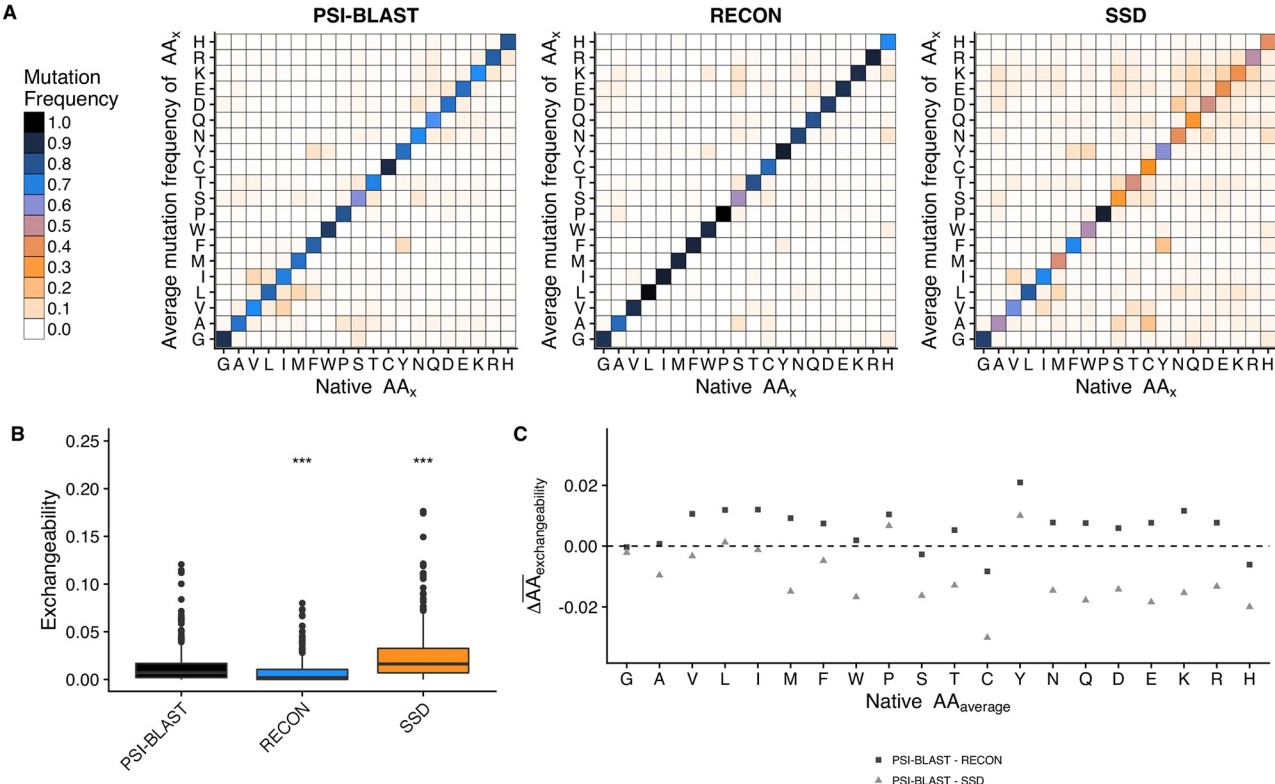

**Fig 4. Comparison of exchangeability rates.** (A) Average amino acid exchangeability of PSI-BLAST, RECON MSD, and SSD sequence profiles. Single-letter amino acid codes were used for both x and y axes, with the x axis representing the original amino acid and the y axis representing the average mutation frequency the original amino acid to the indicated mutation. (B) Comparison of exchangeability rates between profiles, excluding rates of native sequence conservation rates. The y axis represents the mean frequency a native amino acid is replaced with a specific, non-native amino acid, which we term as amino acid exchangeability. (C) Difference of mean amino acid-specific exchangeability observed in a PSI-BLAST profile compared to a design profile. The x axis represents each type of amino acid present in the native sequence. The y axis represents the difference in average exchangeability frequency of each amino acid type, or rather, the average frequency a native amino acid type is replaced with any other non-native amino acid. A positive value indicates the native amino acid is less likely to be exchanged for a non-native amino acid during design, whereas a negative value indicates the native amino acid is more likely to be exchanged, as compared to a PSI-BLAST profile.

## RECON MSD prediction of mutation preferences matches mutation profiles of natural homologues

The application of RECON MSD is not only optimized to engineer a stable, flexible protein, which is also readily accomplished by methods such as consensus sequence design [24]. Rather, RECON MSD explores the sequence space consistent with a protein's flexibility. Thus, we hypothesized, that the sequence space sampled by RECON MSD has similarity to the sequence space sampled in evolution. However, RECON MSD might explore additional sequences which have not yet been explored by natural selection. To answer this question, we compared calmodulin sequence profiles predicted by RECON MSD and SSD to naturally selected calmodulin sequence variation within a curated dataset of calmodulin representative of evolution across all eukaryotes [25]. We also compared the mutation profiles of influenza virus HA2 predicted by either design with mutation profiles obtained from the NCBI Influenza Virus Resource database (IVR) [26]. Given that the HA2 structural models used for this benchmark are of the influenza A H3N2 subtype [27–30], we included all influenza virus type A HA2 sequences deposited in the IVR to generate an HA2 profile. Briefly, in each case, we performed a multiple sequence alignment of all sequences within each database to generate

sequence profiles. From each profile, we measured the root mean square deviation of mutation frequencies to corresponding mutation frequencies predicted by RECON MSD or SSD. For a complete description of the generation of sequence profiles, please consult the Methods section.

The calmodulin and influenza virus type A HA2 mutation frequencies had a root mean standard deviation of 0.473 and 0.580 with respect to RECON MSD profiles and 0.632 and 0.799 to SSD profiles, respectively (Fig 5). Although RECON MSD profiles were more likely to match the mutation tolerances observed within either multiple sequence alignment, the improvement was not uniform for all residues. In general, RECON MSD was more likely to predict matching mutation profiles for residues that undergo local conformational rearrangements, For calmodulin, RECON MSD was more likely to improve the prediction of mutation tolerances within the calmodulin EF-hand at conserved motif positions 3, 5, 9, and 12 [31]. Even though neither ligands nor water molecules were included during protein design, sequence profiles for positions 9 and 12, which are known to provide a bridged water or direct binding to $Ca^{2+}$, respectively, were similar. Within HA2 RECON MSD profiles, residues, charged residues within the B loop, which rearranges into an alpha helix in the post-fusion conformation, and the S5 loop region, which stabilizes the rearranged B loop alpha helix [32], were more likely to have similar predicted mutational tolerances as observed in influenza type A mutation profiles. However, in some cases, RECON MSD residue profiles either failed to improve or even worsened in predicting mutation profile similarity to profiles obtained from multiple sequence alignments as compared to SSD. This was particularly true for positions with small, non-polar residues that maintained contacts within the loosely packed interior within the HA2 trimer, including S93, V100, S113, and I149. Additionally, RECON MSD poorly predicted the conservation of charged residues whose side chains faced the protein interior in at least one conformation of either calmodulin or HA2 trimer, including calmodulin residues Q47, R84, D53, D90, Q132, and D126, and HA2 residue H106.

Because we used only H3N2 backbones to predict HA2 mutation profiles, we subdivided the HA2 profile obtained from all influenza type A multiple sequence alignment into different groups and subtypes including H1 and H2 from group 1 and H3, H4, H7, and H3N2 from group 2 to compare design profile similarity to subtype-specific mutation tolerances (Fig 6, S4 Fig). Separation of HA2 sequences by subtype revealed a divergence in similarity according to related subtypes, with RECON MSD sampling mutation profiles much closer to subtypes within the H3 clade, including H3 and H4. Even so, a Levene's test for equality of variances [33] comparing mutation frequency variance within the H3N2 profile generated by RECON MSD to any influenza subtype A HA2 IVR profile indicated no significant difference between mutation frequency variances within the RECON MSD H3N2 profile and any HA2 IVR subtype profile. This suggests that RECON MSD samples similar mutational tolerances found across naturally selected influenza subtype A mutation tolerances, despite the diverging similarity to group 1 profiles. Given the high sequence profile similarity between RECON MSD and IVR H3 clade profiles, we conclude that RECON MSD can be used to predict possible sequence variation of closely related homologues from a single sequence. This might be particular useful in the case of predicting common mutations that arise due to genetic drift or reassortment, given that the HA2 profile modeled by RECON MSD was not significantly different from multiple HA2 subtype profiles. In cases where RECON MSD substantially deviates from observed mutation frequencies, particularly from the consensus sequence, such mutations warrant further experimental investigation to examine whether these stabilizing mutations within the H3 clade are artifacts of the ROSETTA energy scoring function and/or RECON algorithm sampling, or that they indeed are evolutionary unexplored stabilizing mutations.

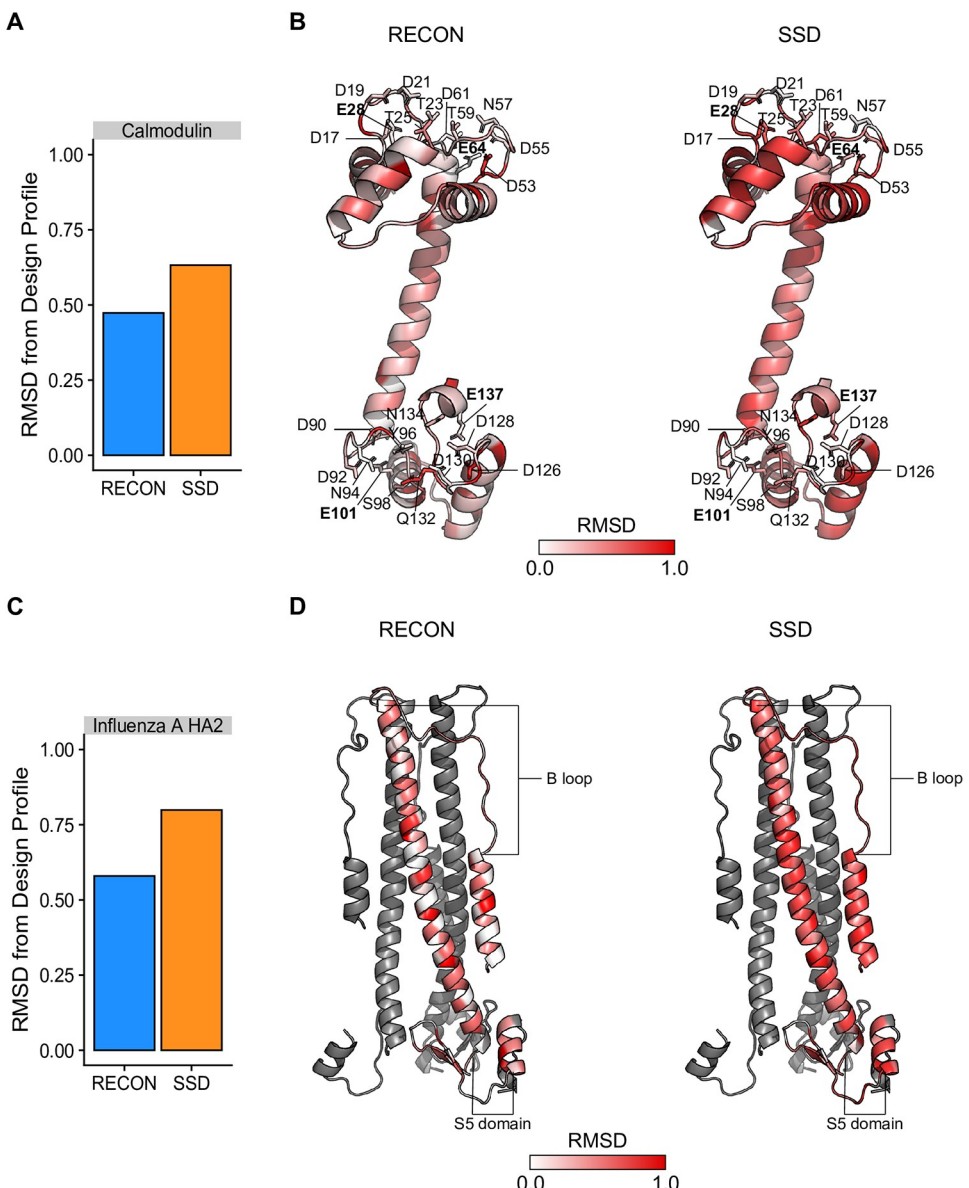

**Fig 5. Comparison of mutation profiles predicted by protein design to mutation profiles observed within calmodulin and influenza type A HA2 multiple sequence alignments.** (A) Comparison of root mean square deviation of mutation frequencies derived from calmodulin natural homologues to mutation profiles predicted by RECON MSD or SSD. Calmodulin natural homologue mutation preferences were derived from the multiple sequence alignment of calmodulin homologues. The root mean square deviation (RMSD) here represents the mean standard deviation of an individual residue's mutation profile, consisting as the mean sum of squared differences of all twenty amino acid frequencies as determined by the multiple sequence alignment of calmodulin homologue sequences in relation to either RECON MSD or SSD residue profile. (B) Residue profile standard deviations between calmodulin multiple sequence alignment profiles and design profiles mapped onto the unbound conformation of calmodulin (PDB ID 1CLL). Here, RMSD represents the mean sum of squared differences of all twenty amino acid frequencies of each residue between homologue and design profiles. Residues whose sequence profiles were predicted to have identical mutation profiles as that within the corresponding position with the multiple sequence alignment are colored in white. The greater the dissimilarity between the homologue mutation profile and design profile, the greater the saturation in red, with complete saturation indicating an RMSD of 1.0. Residues within all four of the conserved EF-hand motifs are labeled, with the bidentate ligand at position 12 critical for Ca$^{2+}$ binding labeled in boldface. (C) Comparison of root mean square deviation of mutation frequencies derived from influenza type A sequence alignments to mutation profiles predicted by RECON MSD or SSD. RMSD is calculated in a similar fashion as in Panel A. (D) Residue profile standard deviations between HA2 multiple sequence alignment profiles and design profiles mapped onto the pre-fusion conformation of the HA2 trimer (PDB ID 2HMG). RMSD is calculated and labeled the

same as in Panel B, but only one HA2 monomer is labeled with RMSD values of the influenza A IVR residue profiles in relation to RECON MSD or SSD profiles. The N- and C-terminal residues of loop regions that undergo large local conformational rearrangements in the post-fusion form are labeled. This includes the B loop that rearranges into an alpha helix and the S5 domain, which stabilizes the alpha helical form of the B loop. Residues within the CR8020 broadly neutralizing epitope [32], including N146 and E150, are also labeled.

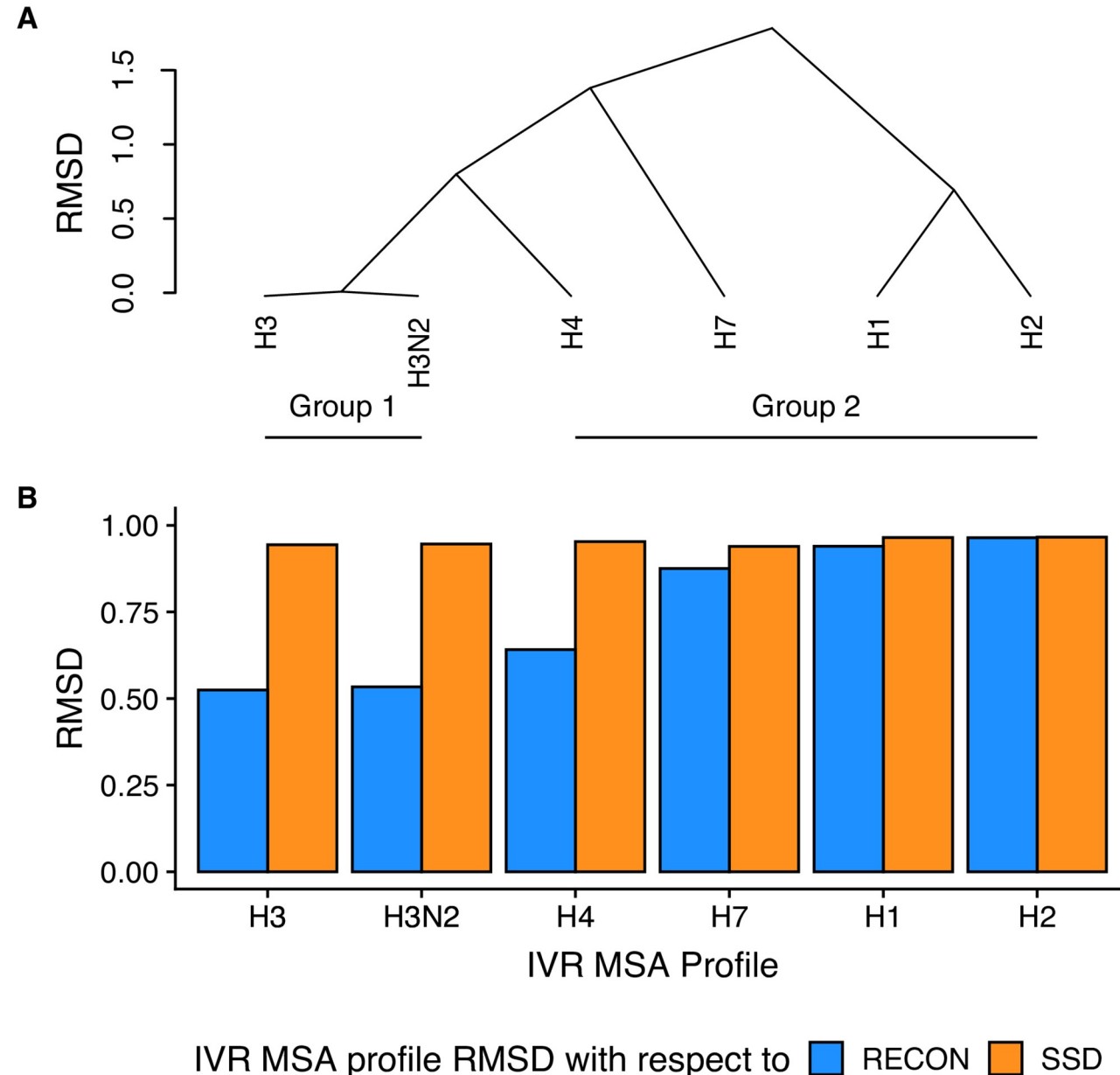

**Fig 6. Root mean square deviations of residue mutation frequencies of influenza A subtypes and HA2 profiles predicted by RECON MSD and SSD.** (A) Dendrogram of root mean square deviations (RMSD) of influenza A subtype HA2 profiles sorted by pairwise RMSD. The mutation frequencies derived from the multiple sequence alignment profile of each influenza A subtype was compared to all other subtypes by calculating the mean standard deviation of each aligned position's mean sum of squared differences of all twenty amino acid frequencies with respect to each other subtype profile. Pairwise RMSD values were sorted to form clades, with the height along the y axis indicating the pairwise RMSD between each clade. (B) RMSD of each IVR subtype multiple sequence alignment (MSA) profile with respect to RECON MSD and SSD. The x axis represents each IVR subtype profile sorted as in Panel A. The y axis represents the RMSD, calculated in the same fashion as in Panel A, of each subtype profile in relation to either RECON MSD or SSD.

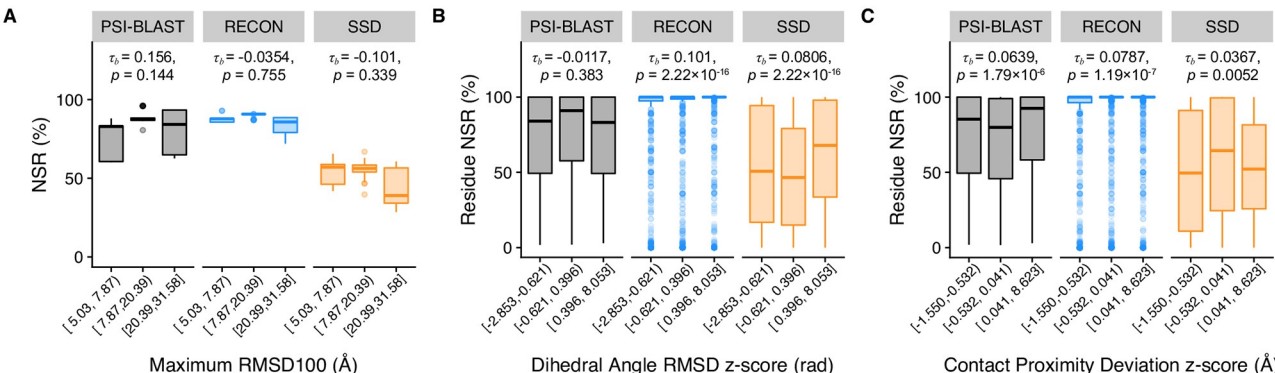

**Fig 7. Relationship of conformational flexibility and native sequence recovery by sequence profiles.** The x axis is binned into three groups of equal number of data points to show the distribution of native sequence recovery between groups of low, middle, and high values for each metric. A Kendall $\tau_\beta$ rank correlation test was performed on each profile to measure the strength of dependence of native sequence recovery on each metric, indicated in each plot along with its associated *p*-value. (A) Comparison of native sequence recovery dependence on maximum RMSD100 between sequence profiles. (B) Comparison of native sequence recovery dependence on $RMSD_{da}$ between sequence profiles. $RMSD_{da}$ values of each protein were not equally distributed, nor were of similar range. Therefore, a *z*-score of was used to normalize $RMSD_{da}$ values of each protein to compare dihedral angle deviation scores, shown along the x axis. A similar approach was implemented to normalize contact map deviation scores. (C) Comparison of native sequence recovery dependence on contact deviation scores.

## Sequence conservation is dependent on its contact map as computed by Cβ–Cβ distance deviations

To consider the effect of conformational flexibility on sequence conservation, we examined the dependency of native sequence recovery on different aspects of conformational flexibility using the aforementioned metrics, maximum RMSD100, $RMSD_{da}$, and contact proximity deviation. We performed a Kendall $\tau_\beta$ rank correlation test on each profile to test for the strength of dependency of native sequence recovery on each metric (Fig 7) [23]. Of the three metrics, the native sequence recovery, or rather percent conservation, observed in PSI-BLAST profiles was only dependent on contact proximity deviation *z*-score, with $p = 1.79 \times 10^{-6}$, versus $p \geq 0.144$ for all other tests. RECON MSD and SSD native sequence recovery depended on both $RMSD_{da}$ and contact proximity deviation *z*-score ($p < 0.01$). Native sequence recoveries of both designed profiles depended strongly on $RMSD_{da}$, with $p \leq 2.22 \times 10^{-16}$, and had similar $\tau_\beta$ coefficients, with $\tau_\beta = 0.101$ for RECON MSD and $\tau_\beta = 0.0806$ for SSD. This finding may suggest that the ROSETTA scoring function employed by both protein design algorithms is too restrictive in sampling for residues at hinge points, given that the same dependency on $RMSD_{da}$ is not observed for PSI-BLAST sequence conservation. In contrast, both PSI-BLAST and RECON MSD had similar $\tau_\beta$ coefficients predicted with the same confidence, with $\tau_\beta = 0.0639$, $p = 1.79 \times 10^{-6}$ and $\tau_\beta = 0.0787$, $p = 1.19 \times 10^{-7}$ respectively, for the dependence of native sequence recovery on contact proximity deviation *z*-score, whereas SSD dis not exhibit the same dependence. This observation suggests that there is an evolutionary constraint on residues that are required to maintain a re-assortment of their local physicochemical environments necessary for a conformational change, and that RECON MSD closely models this evolutionary constraint by considering the multiple local side-chain environments within a protein ensemble.

## Sequences suitable for conformational plasticity are energetically frustrated

The encouraged sequence convergence employed by the RECON MSD algorithm identifies amino acid sequences that have the lowest total energy across all states [12]. To examine the

energetic impact of requiring a single amino acid sequence to adopt multiple states, we use a similar energy score term described previously as the sum total energy score normalized by the number of designed positions (see Methods). For RECON MSD designs, this approach would include lowest mean energy score of the designed ensemble, whereas the SSD energy score would include the lowest energy scores for each state. In all eight cases, RECON MSD selects sequences with a significantly higher energy score than SSD with a paired student's t-test [34], with $p < 1 \times 10^{-4}$ (Fig 8). We also compared the design energy scores to the ten lowest-energy relaxed structures, which only included the native sequences, and found that RECON MSD samples lower energy sequences relative the relaxed native structures. Given that RECON MSD conserves, on average, 88% of native sequences, the few mutations RECON MSD introduces to the native sequence are sufficient to sample a lower energy sequence space than the native sequence. In comparison, SSD sequences are the most stable as SSD replaces the native sequence at a much higher frequency, since SSD optimizes the sequence space for each conformation and can identify much lower energy sequences tolerable for each individual conformation. Therefore, given that RECON MSD is constrained in identifying sequences that are suitable to adopt multiple conformations and that the sequence space identified by RECON MSD is, on average, higher in total energy than the sequence space identified by SSD, it can be inferred that the sequence space available for proteins that populate multiple energy minima is more likely to be energetically frustrated, or at least not as energetically stable, than the sequence space available for a protein which populates a single energy minimum.

### Stability decreases for residues with larger Cβ–Cβ contact map deviations

We used a Kendall $\tau_\beta$ rank correlation test to analyze the dependency of the modeled sequence energy score on global and local conformational changes. For the comparison with global

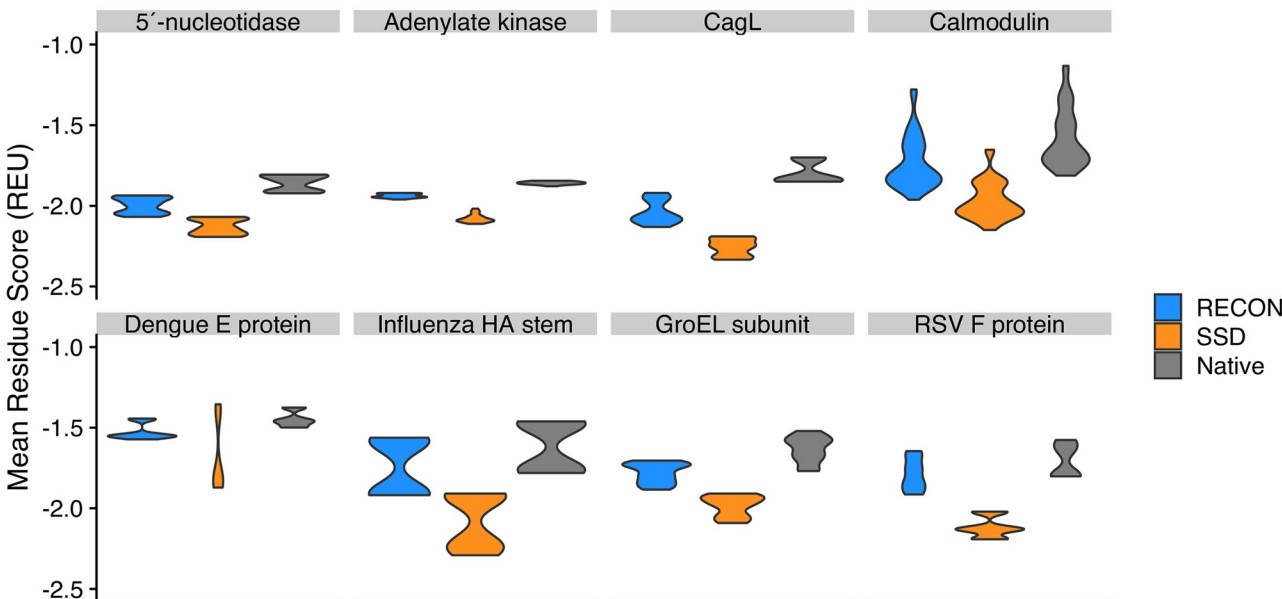

**Fig 8. Average per-residue total energy score of the lowest ten percent scoring models for RECON MSD, SSD, and starting relaxed (Native) models.** One hundred simulations were performed for each group and the lowest ten total energy scoring models were used for the comparison. The total scores were normalized so that the calculated total score was divided by the number of residues within each model to obtain a mean residue score. For RECON MSD models, the total calculated score also had to be normalized by the number of states within each model. The violin plot width indicates the normalized energy score density of each group.

conformational changes, we compared the mean total score of the ten lowest-energy scoring design models, normalized by the number of residues within each protein, to the maximum RMSD100 of an ensemble. We found that there is a negative dependence of mean total score on the maximum RMSD100 for SSD models ($\tau_\beta = -0.143$, $p = 1.16 \times 10^{-5}$), but not so for RECON MSD models ($\tau_\beta = 0.0177$, $p = 0.586$; Fig 9). Conversely, there was a small, but significant positive dependence of individual residue scores on contact proximity deviation $z$-scores for RECON MSD models ($\tau_\beta = 0.0356$, $p = 0.00584$), but not so SSD models ($\tau_\beta = -0.00538$, $p = 0.677$). There was no dependence of individual residue scores on RMSD$_{da}$ for either design approach (Fig 9), which is surprising given that both native sequence recoveries for RECON MSD and SSD were strongly dependent on RMSD$_{da}$. It should be noted that we found the metrices RMSD$_{da}$ and contact proximity deviation were not independent variables, as we determined that contact proximity deviation is significantly, although not strongly, negatively correlated with RMSD$_{da}$ (S5 Fig), meaning that residues with contact proximity deviation values close to or at zero were also more likely to have a higher RMSD$_{da}$ values. In either design case, residues with high RMSD$_{da}$ values tend to have lower scores, *i.e.* score favorably, suggesting that the native contacts and/or hydrogen bonding formed at positions restricted in rearranging side chain proximities for one or all states were more likely to score favorably, or at least more favorably than non-native side chains, by the ROSETTA scoring function [35]. A favorable reference score would prevent the native residue in being redesigned with non-native residues, hence the correlation of high RMSD$_{da}$ values and higher native sequence recovery. Even so, the $\tau_\beta$ correlation of RMSD$_{da}$ and the designed sequences energy was not significant in either design case, such that the degree of backbone flexibility does not directly influence a residue's stability. These data suggest that RECON MSD is restricted in optimizing the stability of residues that must rearrange their local side-chain environments, but not to the same extent in optimizing local backbone flexibility. SSD, on the other hand, is not restricted in optimizing side-chain placement within an ensemble, and therefore can select amino acid sequences that are more stabilizing for individual conformations.

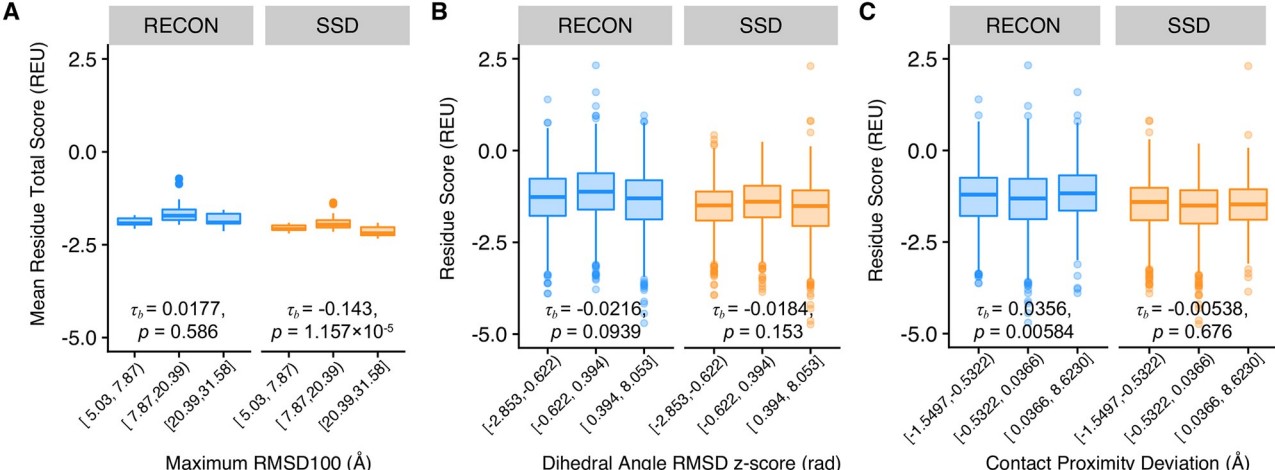

**Fig 9. Comparison of conformational diversity and per-residue total scores.** All panels are binned into low, medium, and high x values, with equal number of data points for each bin. A Kendall $\tau_\beta$ rank correlation test was performed on each profile to measure the strength of dependence of native sequence recovery on the x axis value, indicated in each plot along with its associated *p*-value. (A) Comparison of maximum RMSD100 and mean total energy score, normalized by the number of residues. (B) Comparison of normalized RMSD$_{da}$ *z*-score and mean total energy score of each residue. (C) Comparison of normalized contact proximity deviation *z*-score and mean total energy score of each residue.

# Discussion

## Contact proximity deviation captures local and global conformational rearrangements as a single metric

Methods like Local-Global Alignment and contact area differences are useful in circumventing the over-estimation of global structural dissimilarity by either emphasizing local backbone segment structure similarity or by side chain placement similarity, respectively [36, 37]. In particular, contact area differences between two homologues have been shown to be as accurate as RMSD, if not more, in comparing the structural similarity of proteins with very high sequence similarity [38]. However, contact area differences are sensitive to errors in side chain atom placement within a structural model, so that accounting for side chain mutations while measuring the relative position of equivalent residues is not feasible.

To overcome this limitation, we introduced the contact proximity deviation metric (see Methods section). Contact proximity quantifies the relative placement of a residue within a structure and is sensitive to a change in conformation without relying on side-chain contacts. To decide whether two residues are in contact, we analyze the $C_\beta$–$C_\beta$ distance. However, instead of a hard cutoff distance we use a smooth transition function to avoid discontinuities when distances change by small margins. Contact proximity deviation for a residue becomes the sum of changes in contact proximity when comparing two structures. Therefore, contact proximity deviation quantifies the magnitude of local rearrangements around a residue of interest inflicted by a global conformational change, independent of side chain identity.

Thus, combining the contact proximity deviation metric with interaction network analysis [39] could provide a useful tool to investigate how conformational rearrangements alter residue networks. Additionally, in cases such as the design of protein switches or structural analysis of mutations where protein flexibility, but not sequence, needs to be conserved, it is helpful to have a metric that highlights residues that experience rearrangements in their contacts. Lastly, the measurement of local structural variability by NMR residual dipolar coupling has shown that regions of high flexibility within ubiquitin ensembles align with multiple protein-protein interfaces [40]. Thus, contact proximity deviation provides a possible approach to study protein-protein interaction interfaces.

## Sampling functional mutation preferences requires evaluation of sequence stability as an ensemble

MSD approaches use the energetic contributions of multiple conformations to steer sequence selection and cull any sequences that do not improve the energy score of the designed ensemble as a whole [41]. For most approaches, sequences that do not improve all or the majority of conformations within an ensemble are culled, which is appropriate when the goal is to stabilize a protein ensemble within an energy minimum. However, protein function may not select for sequences that are limited to a single energy minimum. To better estimate the mutational preferences critical for function, it is necessary to use an approach that models local side-chain environments within the context of an ensemble.

Within Fig 4, we illustrated that, although RECON MSD failed to accurately predict all amino acid exchangeability rates, the consideration of multiple local side chain environments during protein design improved the prediction of sequence conservation and overall accuracy of amino acid substitution frequencies as compared to modeling local side chain environments independently, particularly when modeling bulky side chains. In combination with Figs 8 and 9, we demonstrated that the consideration of local side chain stability within the context of an ensemble restricts stability optimization of the ensemble, especially for residues that require

side-chain rearrangements during a conformational rearrangement. Taken together, we demonstrated that selection of mutation profiles by RECON MSD is much more similar to mutation rates observed in homologs if each mutation is evaluated across every conformation, or state, within an ensemble, and then culled if the mutation is evaluated to be destabilizing for an individual state within an ensemble. This approach does not necessarily select sequences that improve the stability of every state within an ensemble, but rather places the importance of modeling an ideal conformation-specific, local side-chain environment to prevent local side-chain destabilization within the context of an ensemble.

## RECON MSD can be used to predict evolutionary sequence conservation of flexible proteins

Sequence similarity searches, such as PSI-BLAST, are fast and easy to use. Predicting mutation preferences from structure, especially if the sequence is known to form multiple conformations, remains to be a challenge. Advances in structure-based evolution design methods rely on iterative approaches that match sequence and structure similarities to predict sequence entropy [42]. For proteins that undergo conformational rearrangements, using this type of approach to search for structural similarity limits the sequence search space to similar conformations, possibly preventing the identification of sequences capable of adopting multiple conformations. Although other MSD methods have improved the selection of more evolutionarily similar sequences, they are limited in their capacity to simultaneously sample conformation and sequence space so that the relevance of conformational plasticity in evolutionary dynamics have not been fully interrogated.

The caveat to using RECON MSD to predict mutation preferences is accounting for ROSETTA sampling biases. First, RECON MSD does not currently allow for the formation or destruction of disulfide bonds, which is critical for conformation stability, and does not accurately model the frequency of cysteine conservation. Consideration of alternate protonation states due changes in pH are also not explicitly modeled, which we see from our amino acid exchangeability comparisons that RECON MSD underrepresents exchangeability of polar residues and frequently mutates histidine to lysine or arginine, which has a $pK_a$ much higher than histidine or which is not charged. Additionally, in Fig 6, we showed that RECON MSD is likely to overestimate sequence conservation of hinge regions that have large dihedral angle RMSDs. Even though we used a gentle minimization prior to design, minimization significantly increases the estimated stability of the native residue, making the replacement of the native amino acid unfavorable, as shown in S1 Fig. Given that residues located at hinge points within flexible loops are intrinsically disordered and typically contain less than ideal Ramachandran dihedral angles, it is likely that minimization specifically overcorrects these bond angles to fit the energy scoring function, preventing accurate sampling of rotamer placement. With the addition of explicit disulfide bond formation, use of a $pK_a$-dependent rotamer library, and improvement of minimization prior to design, the RECON MSD algorithm could prove to be a valuable tool in predicting accurate mutation profiles.

With that being said, we used RECON MSD to demonstrate that sequence conservation and mutation preferences of a single sequence can be approximated using the evaluation of local residue physicochemical changes, provided that this one sequence folds into select, multiple conformations. In Fig 3, we showed that the estimated sequence conservation of RECON MSD designs differs by roughly 5% from the sequence conservation observed in PSI-BLAST profiles, with RECON MSD being more conservative. More specifically, we demonstrated that RECON MSD samples a very similar sequence space for hemagglutinin (HA2) compared to what has been observed in H3 clade influenza subtypes (Fig 6). Being able to predict the

tolerated sequence space for viral antigens such as influenza HA has possible applications for antiviral drug design. It would have been preferred to compare the design profiles to deep sequencing data, as the represented mutation frequencies within the IVR database likely underestimate rare mutations. However, given the correspondence of the RECON MSD predicted HA2 sequence profiles and IVR subtype-specific sequence profiles, it stands to reason that RECON MSD can serve an *in silico* approximation for costly deep sequencing, or at least serves as an initial screening for potential, more frequently observed mutations of drug targets, such as pathogens or oncoproteins.

The computational time required for the RECON MSD design simulations within this benchmark ranged from 2–36 hours. Compared to experimental approaches that have tested for functionally tolerated mutations in either dengue virus envelope protein or influenza hemagglutinin protein [2, 43, 44], RECON MSD is much faster and less costly in identifying biologically relevant mutations. Additionally, RECON MSD is not limited to sampling mutations singly, pairwise, or as limited networks, but rather can sample mutations as an interaction network of each local side-chain environment. Traditional intra-protein co-evolution methods, such as direct coupling analysis [45], mutual information [46–48], or McLachlan-based substitution correlation methods [49, 50], are not reliable in detecting co-variation or correlation of mutation frequencies of highly conserved sequences [51], and so they fail to detect contact dependencies of sequences with low sequence variation. In the case of this benchmark, we see that flexible sequences tend to be more highly conserved, especially when residues need to maintain distinct contacts between conformations. Therefore, current co-evolution methods cannot be used to detect residue contact dependencies of flexible, highly conserved sequences, whereas this benchmark suggests that RECON MSD is well-suited to identifying the evolutionary potential of a flexible sequence.

## Conclusions

We demonstrated that RECON MSD significantly improves the similarity to evolutionary mutation preferences from SSD selected mutation profiles by selecting sequences which are energetically favorable for an ensemble of local side-chain interactions. Specifically, in instances where the goal of protein design is to preserve an ensemble of conformations for functionality, we suggest a greater emphasis on designing local physicochemical environments for each and all conformations within an ensemble, and to place less of an emphasis of finding sequences representing the most thermostabilizing for either each state individually or as an average of all states. Furthermore, the new conformational diversity metric contact proximity deviation we describe in this paper allows for the comparison of protein ensembles, assuming they are of similar length but not sequence, by quantifying position-specific relocation due to one or more conformational changes. Therefore, in conjunction with contact proximity deviation, RECON MSD warrants further use as a bioinformatic tool to estimate mutation preferences of homologous proteins, especially for proteins known to undergo similar domain or fold reorganization between conformations.

## Methods

### Selection and preparation of benchmark datasets

Our criteria for benchmark datasets included proteins that had at least two published conformations with greater than 5 Å RMSD and at least one peptide chain greater than 100 residues in length. To identify these proteins, we performed a BLAST search to identify proteins with 100% sequence identity and with gaps of three or less residues in length. Structures with

similar backbone conformations of less than 0.5 Å RMSD were excluded from design so that the structure with the longest matching consecutive sequence was kept as the template structure.

Structures were downloaded from the Protein Data Bank (PDB; www.rcsb.org) and processed manually to remove all atoms other than the residue atoms intended for design. Any residues that did not align or positions that were not present in all template structures were not considered for design and were removed from the template. For a detailed description of which residues were included for design, see S1 Table. Native structures were subject to minimization and repacking in ROSETTA using FastRelax constrained to start coordinates with the talaris 2013 score function placing a backbone movement constraint on all $C_\alpha$ atoms of 0.5 Å standard deviations to prevent substantial movement away from the native structure [35, 52]. The lowest total energy score model was selected from the 100 relaxed models for design. For comparisons using un-relaxed models, the native structure was used instead of the relaxed model.

## RECON multi-state and single-state design

Benchmarking using RECON MSD design was performed using four rounds of fixed backbone design and a convergence step using the greedy selection algorithm, as previously described [12, 13], with the exception that only repacking, and not backbone minimization was allowed following the convergence step to prevent over-optimization of the energy score following design. For parallelized production runs of multistate design, each state within an ensemble was handled on its own processor, requiring up to 32GB of RAM per node for the largest design case, RSV F trimer. Similarly, single-state design was performed using four rounds of fixed backbone rotamer optimization followed by repacking using the identical designable residues as specified for RECON MSD designs. The talaris 2013 scoring function was used for both RECON MSD and single-state designs [35]. One hundred designs were generated for each benchmark structure using either RECON MSD or SSD.

## Generation of sequence profiles

The lowest ten out of a hundred scoring models were used for quantification of sequence tolerance. In the case of SSD, the ten lowest total scoring models were used from the design simulation of each PDB structure and then were grouped by protein to form an ensemble containing $10 \times N$ models, with $N$ being the number of conformations within an ensemble. For RECON MSD, the total score of each model designed within an ensemble design run was averaged with all other conformations modeled during the same RECON MSD run to create a fitness score, which then was sorted to identify the ten designed ensembles with the lowest fitness score, again containing $10 \times N$ models for each ensemble. A Shannon entropy bitscore was calculated using WebLogo [53] for each designed position within an ensemble as $I_i = p_i \times \log_2(20 \times p_i)$, with $i$ as the amino acid and $p_i$ as the frequency of that amino acid. Here, the calculated amino acid frequency includes the frequency at the same position within the ten lowest-scoring models of all designed states, whether designed independently by SSD or designed simultaneously by RECON MSD, such that an amino acid represented a 100% of the time at a particular position in all states has a bitscore of 4.32 [54]. The frequency of each possible mutation, *i.e.* all twenty amino acid frequencies recovered from design at each position, was calculated from bitscores of each position to generate a $20 \times n$ matrix, with $n$ being the number of designed residues within each ensemble.

PSI-BLAST profiles were obtained by querying a non-redundant protein database using default parameters, increasing the number of iterations to ten iterations, as well as querying

the database with $e$-value thresholds ranging from $1 \times 10^{-5}$ to $1 \times 10^{2}$. We reported only PSI-BLAST profiles generated using default parameters, which includes two iterations and an $e$-value of 0.005. PSI-BLAST profiles using non-default parameters were qualitatively identical to the PSI-BLAST profiles generated using the default parameters, and were therefore not reported. We omitted any sequences within the queried sequence profiles which were not included for design to generate a $20 \times n$ matrix corresponding to each benchmark case containing the amino acid frequencies obtained from the position specific-scoring matrix (PSSM) as described in the previous paragraph.

### Methods for comparison of sequence profiles

We compared the PSSM generated from the PSI-BLAST query to the PSSM generated by either RECON MSD or SSD by calculating the percentage of native sequence recovery, which was determined as the sum of the bitscores of the native amino acids at each position divided by the sum of the information bitscore of all amino acids at all positions [55]. Additionally, we calculated the root mean square deviation in designed position mutation frequencies obtained from the bitscores of each position with the PSSM, as described in Fig 3:

$$y = \sqrt{\frac{\sum_{i=1}^{n} \sum_{j=1}^{20} \left(aa_{PSI-BLAST} - aa_{Design}\right)^2}{n}}$$

where $aa_j$ represents the frequency of an amino acid observed at position $i$ for each of all twenty amino acids ($j$), and $y$ is the sum of all $i$ differences for all amino acids within a protein with a length of $n$ residues [56]. Average amino acid substitution rates were determined as the mean cumulative substitution frequency of each amino acid $i$ to amino acid $j$, where

$$\overline{aa_{i,j}} = \begin{bmatrix} \frac{1}{aa_1} \sum aa_{1,1} & \cdots & \frac{1}{aa_1} \sum aa_{1,j} \\ \vdots & \ddots & \vdots \\ \frac{1}{aa_i} \sum aa_{i,1} & \cdots & \frac{1}{aa_i} \sum aa_{i,j} \end{bmatrix}.$$

We define amino acid exchangeability as the subset of average amino acid substation rates that exclude the substitution rates of $i \rightarrow j$, where $j$ is identical to $i$, or in other words, all substitution rates that include the average conservation frequencies of the native amino acid [57]. The mean amino acid exchangeability rates, as shown in Fig 3C, were calculated as

$$\overline{aa_i} = \frac{1}{19} \sum_{j=1}^{19} \frac{1}{aa_i} \sum aa_{i,j}$$

where the mean exchangeability rate of each amino acid $i$ is the mean of all exchangeability rates of amino acid $i$ to amino acid $j$, excluding the conservation rate of amino acid $i$. Kendall $\tau_\beta$ rank comparison tests, linear regression, Wilcox comparison of means, and student t tests were performed in R. Levene's test for equality of variance was performed using Python 3.7 using the scipy.stats package using the median as the center.

### Preparation of calmodulin and influenza type A HA2 mutation profiles and consensus sequences

The entire influenza type A HA sequences were obtained from the IVR using the following parameters–type: A, Host: any, Country/Region: any, Protein: HA, Subtype H: any, Subtype N:

**Table 2. Number of sequences within each influenza type A HA sequence dataset.**

| Influenza Type A subtype | Number of Sequences in IVR |
| --- | --- |
| All | 73943 |
| H3N2 | 29698 |
| H3 | 32645 |
| H4 | 1823 |
| H7 | 2424 |
| H1 | 23688 |
| H2 | 608 |

any. Subdivision of the entire influenza type A HA sequences were obtained by changing the parameter Subtype H to 1, 2, 3, 4, or 7 with Subtype N as any, or for the H3N2 subtype, Subtype H: 3 and Subtype N:2. The number sequences obtained from each query are listed in Table 2. Sequences within the calmodulin dataset containing non-redundant calmodulin sequences across eukaryotes obtained from the Supplementary Material in [25] or influenza type A full-length HA sequences obtained from the Influenza Virus Resource [26] were aligned using a locally installed Clustal Omega version 1.2.4 [58, 59]. Given the limited number of sequence gaps and high sequence conservation, we believe the multiple sequence alignments performed on either dataset were accurate. The frequency of each amino acid type at each aligned position was determined using WebLogo [53] using the following parameters–sequence-type: protein, format: logodata, composition: none.

## Description of conformational metrics used in this benchmark

We use the maximum RMSD within an ensemble to represent the largest amplitude of dissimilarity within an ensemble, defined as:

$$RMSD_{max} = max_s \sqrt{\frac{1}{n} \sum_{i=1}^{n} \|v_i - w_i\|^2}$$

where $n$ represents the number of residues, $s$ represents the number of structures within an ensemble, and $\sqrt{\frac{1}{n} \sum_{i=1}^{n} \|v_i - w_i\|^2}$ represents each pairwise RMSD within an ensemble (Fig 2A) [20]. For the local backbone dissimilarity metric, we use dihedral angle RMSD to describe the deviation of each equivalent dihedral angle, or pair $\phi$ and $\varphi$ angles, within an ensemble containing $s$ structures, as

$$RMSD_{da} = \sqrt{\frac{1}{s} \sum_{i=1}^{s} \|\phi_s - \bar{\phi}\|^2 + \frac{1}{s} \sum_{i=1}^{s} \|\psi_s - \bar{\psi}\|^2}.$$

where $-\phi$ and $-\psi$ represent the mean $\phi$ and $\psi$ angle of each equivalent residue. The contact map dissimilarity metric we introduce here is based off the Durham et al. [60] neighbor count weight metric, which scores the likelihood of a neighboring contact by assigning each $C_\beta$–$C_\beta$

distance a score, which we term contact proximity as

$$
contact\ proximity
$$
$$
= \begin{cases} 1, & if\ distance \leq lower\ bound \\ \frac{1}{2}\cos\left(\frac{distance - lower\ bound}{upper\ bound - lower\ bound}\right), & if\ lower\ bound < distance < upper\ bound \\ 0, & if\ distance \geq upper\ bound \end{cases}
$$

For glycines, a pseudo-$C_\beta$ atom is defined using the amide N, $C_\alpha$, and carboxyl C atom coordinates in the PDBtools package (github.com/harmslab/pdbtools) before calculating $C_\beta$–$C_\beta$ distances. The lower and upper bounds represent thresholds where a $C_\beta$–$C_\beta$ distance certainly does and does not contain any side-chain atoms that are in contact with another residue's side-chain atom. We define the lower bound as 4.0 Å and the upper bound 12.8 Å, where the lower bound was determined to be a reliable threshold to define solvent-inaccessible side-chains due to side-chain contacts, or in other words, a $C_\beta$–$C_\beta$ distance less than the lower bound is very likely to form at least one side chain interaction [60]. The upper bound was determined by the maximum $C_\beta$–$C_\beta$ distance where at least one atom from each side chain formed an interaction [61]. Finally, the contact proximity deviation for each residue was calculated as the sum of all $C_\beta$–$C_\beta$ contact proximity score deviations for that residue (Fig 2D). With this metric, we can quantify the changes in side chain local environments that are not due to local hinge-points, but instead, show local side chain environment changes that are due to larger conformational rearrangements.

## Supporting information

**S1 Table. Residue number and starting sequence of protein benchmark set used for design.** The chain and residue numbers for each PDB along each row were designed separately, except for proteins with multiple chains included in design, such as influenza virus HA2 and RSV F, where a single chain and corresponding residue numbers are included on an individual line. A grey line that marks the end of all chain and residue numbers included for one PDB. The notation ([]) denotes chain breaks within the sequence. The (*) indicates that gaps in the alignment where there was incomplete density in the crystal structure. For input PDB models, including 1OK8, 3C5X and 3C6E, the sequence from 3J27 was threaded onto the missing densities in structures 1OK8, 3C5X, and 3C6E so that there were no gaps. A detailed description of the preparation of input models for design is included in the S1 Appendix.
(DOCX)

**S1 Appendix. Protocol capture.** The following document includes a detailed description of model preparation, protein design, and analysis methods used in this manuscript, including the software versions and command line options. Command line options are written in monospace. The '\\' symbol when included in command line options indicates a wrapped single line. Scripts requiring either a Python or R environment are indicated.
(PDF)

**S1 Fig. Design native sequence recovery and mutation profile variability comparisons to PSI-BLAST profiles using relaxed and unminimized starting models.** (A) Comparison of total native sequence recovery of relaxed and unminimized RECON MSD and SSD designs to PSI-BLAST sequence profiles generated using the native sequence. Asterisks indicate the significance of difference of means of each design in comparison to the PSI-BLAST profile, with a $z$-test $p$-value $< 0.01$ represented by one asterisk, and a $p$-value $< 0.00001$ by three asterisks.

(B) Mutation frequency variances of designs in comparison to a PSI-BLAST profile, normalized by protein length. The y-axis values represent the average variability of mutation profiles for each designed residue in relation to a PSI-BLAST profile, as described in Fig 3.
(TIF)

**S2 Fig. Difference in average amino acid exchangeability between sequence profiles.** The x axis represents the original amino acid. The y axis represents the difference in average mutation frequencies between two profiles, which are noted above each grid. Along the diagonal axis, indicating native sequence conservation, values less than zero (oranges) signify that the latter profile was more highly conserved, and values greater than zero (blues to black) signify that the native residue was less conserved in the latter profile. Not along the diagonal, values less than zero indicate that exchangeability of the native residue to the indicated residue along the y axis was higher in the latter profile, whereas values greater than zero indicate that exchangeability was lower in the latter profile.
(TIF)

**S3 Fig. Comparison of individual exchangeability rates.** (A) Scatterplots of each exchangeability rate as observed in a PSI-BLAST profile compared to design profile. Both the x and y axes represent the exchangeability frequency of a native amino acid to a specific, non-native amino acid, with PSI-BLAST exchangeability rates along the x axis, and design exchangeability rates along the y axis. For reference, a grey line drawn is drawn along where the exchangeability rates would be equal between a PSI-BLAST and design profile. Both an adjusted $r^2$ and $\tau_\beta$ value is provided, along with the associated two-sided $p$-value. Lighter points are found along the linear regression model, and darker points represent outliers. (B) Measures of influence for individual exchangeability rate. The index listed along the x axis refers to each exchangeability rate, indexed in order alphabetically. For reference, in the first nineteen indices, the first index refers to the A to C mutation frequency, followed by the next eighteen indices that correspond with A to D through Y mutation frequencies. The measure of influential observation, or DFBETA index, is represented along the y axis. The height and direction of each bar corresponds with the change in regression model correlation coefficient without that particular observation. Influential outliers that have $> \left| \frac{2}{\sqrt{N}} \right|$ index value, or ±0.106 threshold, are colored in black and are labeled with the associated mutation.
(TIF)

**S4 Fig. Root mean square deviation of residue mutation preferences between influenza A subtype multiple sequence alignments and their RECON MSD and SSD profiles.** Each IVR subtype mutation profile was generated by multiple sequence alignment of HA2 sequences within the IVR database, subdivided by HA subtype including H1, H2, H3, H4, and H7. Because the designed sequence used only an H3N2 HA2 backbone, the H3N2 subtype was included in addition to H3. Only positions that align to the native sequence used for design were included within the profile. HA2 subsequences are separated and ordered by similarity to H3N2, from highest similarity on the top. The x axis each aligned position of the HA2 sequence, corresponding to the H3N2 residue numbering of PDB ID 2HMG, chain F. The y axis is the root mean square deviation (RMSD) of each residue's subtype-specific profile within the multiple sequence alignment with respect to RECON MSD, on the left, and to SSD on the right.
(TIF)

**S5 Fig. Correlation of dihedral angle RMSD and Cβ−Cβ distance deviation.** (A) The x-axis represents dihedral RMSD, measured in radians, and the y-axis represents contact proximity deviation, measured in Å. The hex bins shaded in grey are the number of residues within the

deposited PDB structure have have both a $C_\beta$–$C_\beta$ distance deviation and dihedral angle RMSD within a bin. (B) Axes represent same metrices as in Panel A, normalized by z-score. (TIF)

## Acknowledgments

The authors would like to thank Amanda Duran, Axel Fischer, and Diego del Alamo for their assistance and helpful discussions during the development of this project. Calculations for the contact map deviations were made possible by members of the Michael Harms Laboratory as contributors to the PDBtools opensource software (github.com/harmslab/pdbtools). We also thank the Vanderbilt University Advanced Computing Center for Research and Education for the computational resources necessary to complete this project.

## Author Contributions

**Conceptualization:** Marion F. Sauer.

**Data curation:** Marion F. Sauer.

**Formal analysis:** Marion F. Sauer.

**Funding acquisition:** James E. Crowe, Jr., Jens Meiler.

**Investigation:** Marion F. Sauer, James E. Crowe, Jr., Jens Meiler.

**Methodology:** Marion F. Sauer, Jens Meiler.

**Project administration:** Jens Meiler.

**Resources:** Jens Meiler.

**Software:** Marion F. Sauer, Alexander M. Sevy.

**Supervision:** James E. Crowe, Jr., Jens Meiler.

**Validation:** Marion F. Sauer.

**Visualization:** Marion F. Sauer.

**Writing – original draft:** Marion F. Sauer.

**Writing – review & editing:** Marion F. Sauer, Alexander M. Sevy, James E. Crowe, Jr., Jens Meiler.

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
