## [Decision Letter · Decision Letter 0]

16 Sep 2019

Dear Dr Sauer,

Thank you very much for submitting your manuscript 'Multi-State Design of Flexible Proteins Predicts Sequences Optimal for Conformational Change' for review by PLOS Computational Biology. Your manuscript has been fully evaluated by the PLOS Computational Biology editorial team and in this case also by independent peer reviewers. The reviewers appreciated the attention to an important problem, but raised some concerns about the manuscript as it currently stands. While your manuscript cannot be accepted in its present form, we are willing to consider a revised version in which the issues raised by the reviewers have been adequately addressed. We cannot, of course, promise publication at that time.

If you have any concerns or questions, please do not hesitate to contact us.

Sincerely,

Björn Wallner

Associate Editor

PLOS Computational Biology

Arne Elofsson

Deputy Editor

PLOS Computational Biology

[LINK]

Reviewer's Responses to Questions

**Comments to the Authors:**

Reviewer #1: The Meiler lab previously described their method for multistate design called RECON and have used it recently in improving the cross-reactivity of an anti-influenza antibody. Very briefly, RECON is more efficient than conventional multistate design algorithms because it uses a heuristic that first samples sequence space across all possible protein states and then defines sequence constraints based on this sampling to encourage the convergence of all states to a single sequence acceptable to all. Here, Sauer et al. benchmark the method's ability to produce sequences of conformationally flexible proteins that are similar to the sequence space of their natural homologs. This problem had been tackled previously as the authors note, for instance by Kortemme's lab, and yet, it is clear that the method used here is, at least in principle, superior to those used previously. Briefly, past applications of multistate design to this problem looked for sequence profiles that are "on average" acceptable across all states independently, whereas here, the sequences are actually tested on each and every state. The authors also describe a new metric for estimating conformational differences between protein states. Although this metric may be novel, they do not present a compelling case for its usefulness.

Major comment:

* Throughout the paper, the authors point out that RECON is more conservative in mutations than alternative single-state design methods. This makes a lot of sense and is expected. The problem is that the benchmark provided in the paper -- the designs' compatibility with the natural homologs' sequence diversity -- may therefore be misleading. The most conservative design method, i.e., consensus design, would probably be even superior to RECON according to this metric. The main advantage of a design algorithm is obviously if it can come up with sequence choices that are non-trivial, that is, they don't appear in the natural diversity and yet are compatible with function. Given that there are now large deep mutational scanning datasets for many proteins, it seems natural to test whether RECON "predicts" these data. I hardly expect a perfect prediction (this is a very tough challenge), but an improvement relative to PSI-BLAST and single-state design methods is expected. Of course, successful prediction in this case would also make the claim of usefulness much stronger as these deep sequencing analyses are expensive and time-consuming.

Minor comments:

In parts, the paper lacks clarity and some of the suggestions below are meant to clarify:

* The abstract states an increase of 30-40% in sequence conservation, but it is not clear how this is measured at this point. Either rephrase more accurately or eliminate the details here. Also, "RECON designs resembled the evolutionary sequence space of functional proteins" is both vague and as stated in Major comment above, largely to be expected for a conservative method.

* I recommend to eliminate the first sentence of the introduction. It's far too broad for the current paper.

* The letter to the editor mentions that the method may lead to anticipating escape mutations in pathogens. I was intrigued by this suggestion but could find no mention for it in the paper. If the authors think this is realistic, they should explain in the Introduction/Discussion or provide some suggestive data.

* line 64: "we hypothesize": it's trivial to assume that functionally essential flexibility limits sequence space. The word hypothesize is an overstatement. In particular, the authors cite other papers that already made this claim.

* As the introduction is currently written, to understand the current contribution properly, the reader needs to have a fairly in-depth acquaintance with RECON and its predecessors, particularly the one from the Kortemme lab. I suggest to explain more deeply both of these methods to provide the necessary detail.

*line 100: "select the lowest-energy sequence": ill-defined. MSD methods select a sequence that finds a compromise among all states, not the lowest-energy sequence.

* lines 245-246 explain the sequence-recovery benchmark. This is critical for understanding the results but I couldn't make sense of this explanation and I therefore only have a vague idea.

* similarly in lines 257-261

Reviewer #2: The main objective of protein design is to determine a favorable amino acid sequence for a given 3D structure. But this problem becomes quickly intractable due to the large amino acid sequence space from which a suitable sequence must be identified. Therefore, state-of-the-art methods make use of heuristic algorithms to explore the sequence space. Moreover, it is evident that these methods have demonstrated success by designing proteins de novo for developing therapies. In most of these studies, the authors have sought an amino acid sequence for the most energetically stable protein structure or a single state conformation. However, these methods are not necessarily helpful for designing a class of (flexible) proteins that exist in multiple conformational states or an ensemble. The complexity of such design problem increases significantly even for small flexible proteins. RECON (REstrained CONvergence) is an efficient method that can design sequences for an ensemble of conformations of large flexible proteins as shown in studies published in 2015 and 2019.

In this study, Sauer and colleagues utilize RECON to perform multi-state design (MSD) of large flexible proteins and demonstrate that the sequence profiles predicted favor conformational change. In particular, the designed sequences resemble evolutionary sequence profiles and tolerate mutations across all the conformations of an ensemble. Here, the authors compare sequences designed for a benchmark set of eight proteins, using RECON MSD with (i) sequences designed for each single conformation (single-state design or SSD) of an ensemble, and (ii) PSI-BLAST sequence profiles. The success of RECON MSD is obvious from its native sequence recovery which is 88% compared to 49% recovery from SSD or 82% from PSI-BLAST on average. Through alternative tests, authors show that RECON MSD doesn’t favor mutations as much as SSD, but the preferred mutations are close to those observed in evolution. While sequences sampled from RECON MSD are not as stable energetically as SSD sequences, they recognize mutations tolerated within the native structures. Here, the authors also establish a new metric called contact proximity deviation, that can be used to quantify changes in contact maps that capture the magnitude of local and global conformational flexibility across a structural ensemble. Later, they show that sequence conservation in the flexible regions depends heavily on the contact proximity deviation score (Fig 5). Through RECON MSD and contact proximity deviation score, the authors further demonstrate that they are able to predict sequences that are stable within the context of an ensemble but not necessarily for individual conformations when considered separately, thereby taking into account changes in local side-chain environment in the ensemble. The authors also present current limitations of the RECON MSD such as sampling bias from Rosetta and overestimation of sequence conservation in the hinge regions. This method is efficient given that it took up to 36 hours for proteins up to 1300 residues.

In summary, the current manuscript by Sauer et al demonstrates that RECON MSD can be utilized to design evolutionary sequence profiles of large flexible proteins. This can be very useful in understanding tolerated mutations in native proteins that are not yet revealed by evolution. Furthermore, the authors introduce a new metric that measures local conformational flexibility that occurs due to large structural changes. In general, the manuscript is very well-written and presents data in a logical and sound manner. 

Minor Comments:

1. In the section “RECON MSD samples sequence profiles that are more similar to evolutionary observed sequence profiles when compared to SSD”, it is demonstrated that SSD favors mutation over RECON MSD. But, from section “Regions critical for conformational plasticity are energetically frustrated”, it seems like the opposite is true. These seem like they are contradicting one another. Can you please clarify?

2. References were not found in lines: 408, 623, 633, 647.

3. Figure S4 is not referenced anywhere (or maybe it is one of the references that is missing as pointed out in my previous comment). Can you please elaborate on this? I think this should further clarify why RECON MSD sequence conservation is dependent on both RMSDda and contact map deviation shown in Figure 5.

**Have all data underlying the figures and results presented in the manuscript been provided?**

Reviewer #1: Yes

Reviewer #2: No: Fig. S4.

PLOS authors have the option to publish the peer review history of their article (what does this mean?). If published, this will include your full peer review and any attached files.

Reviewer #1: No

Reviewer #2: No

---

## [Decision Letter · Decision Letter 1]

23 Dec 2019

Dear Dr Sauer,

We are pleased to inform you that your manuscript 'Multi-State Design of Flexible Proteins Predicts Sequences Optimal for Conformational Change' has been provisionally accepted for publication in PLOS Computational Biology.

In the meantime, please log into Editorial Manager at https://www.editorialmanager.com/pcompbiol/, click the "Update My Information" link at the top of the page, and update your user information to ensure an efficient production and billing process.

One of the goals of PLOS is to make science accessible to educators and the public. PLOS staff issue occasional press releases and make early versions of PLOS Computational Biology articles available to science writers and journalists. PLOS staff also collaborate with Communication and Public Information Offices and would be happy to work with the relevant people at your institution or funding agency. If your institution or funding agency is interested in promoting your findings, please ask them to coordinate their releases with PLOS (contact ploscompbiol@plos.org).

Thank you again for supporting Open Access publishing. We look forward to publishing your paper in PLOS Computational Biology.

Sincerely,

Björn Wallner

Associate Editor

PLOS Computational Biology

Arne Elofsson

Deputy Editor

PLOS Computational Biology

Reviewer's Responses to Questions

**Comments to the Authors:**

Reviewer #1: I'd like to commend the authors for making such an effort to address the reviewers' comments. I think that the paper is clearer and deeper now and I have no further comments.

**Have all data underlying the figures and results presented in the manuscript been provided?**

Reviewer #1: Yes

PLOS authors have the option to publish the peer review history of their article (what does this mean?). If published, this will include your full peer review and any attached files.

Reviewer #1: Yes: Sarel Fleishman

---

## [Editor Report · Acceptance letter]

23 Jan 2020

PCOMPBIOL-D-19-01368R1 

Multi-State Design of Flexible Proteins Predicts Sequences Optimal for Conformational Change

Dear Dr Sauer,

I am pleased to inform you that your manuscript has been formally accepted for publication in PLOS Computational Biology. Your manuscript is now with our production department and you will be notified of the publication date in due course.

With kind regards,

Matt Lyles
